# Machine learning-supported framework for the classification of mpox infection and MVA immunization from multiplexed serology data

Rebecca Surtees[1,8], Fridolin Treindl[2,8], Shakhnaz Akhmedova [3,8], Denis Beslic [3,8], Fatimanur Bayram [1], Akin Sesver [1], My Linh Nguyen[1], Thomas Rinner[1], Marica Grossegesse [1], Martin Skiba[2], Janine Michel [1], Nils Körber [3], Klaus Jansen [4], Uwe Koppe [4], Ulrich Marcus [4], Nicole Friedrich [5], Annette Mankertz [5], Katharina Ladewig [6,7], Hans Werner Mages[2], Brigitte G. Dorner [2], Andreas Nitsche [1] ✉ & Daniel Stern [2] ✉

The 2022 global mpox outbreak highlighted the risk of zoonotic diseases establishing sustained transmission in human populations and underscored the need for accurate serological tools to monitor orthopoxvirus exposure. However, cross-reactive antibodies induced by Modified Vaccinia Ankara (MVA) vaccination make it difficult to discriminate between monkeypox virus (MPXV) infection and vaccination-induced immunity. Here we present a machine learning (ML)-assisted bead-based serological multiplex assay that distinguishes MPXV infection from MVA vaccination and pre-immune sera by targeting antibody responses to 15 poxviral antigens. Of the six algorithms tested, the Gradient Boosting Classifier (GBC) achieves the highest performance (F1 = 0.83) in sera from the 2022 outbreak and from a follow-up epidemiological cohort of at-risk men who have sex with men (MSM; n = 1,260). In an independent validation cohort (n = 143), GBC (F1 = 0.70) robustly detects MPXV infections, including breakthrough cases, with 88% specificity and 92% sensitivity. Integrating ML with high-dimensional serology enables accurate cross-sectional classification of orthopoxvirus immune status and provides a scalable framework for mpox serosurveillance and outbreak preparedness.

Mpox is caused by infection with monkeypox virus (MPXV), a member of the *Orthopoxvirus* genus within the *Poxviridae* family. While variola virus, the causative agent of smallpox, was declared eradicated by the WHO in 1980, zoonotic transmissions of other orthopoxviruses, including MPXV, cowpox virus (CPXV), and vaccinia virus (VACV), have increased globally in recent years[1]. This resurgence is largely attributed to the growing proportion of the population that is susceptible to orthopoxvirus infections, following the global discontinuation of smallpox vaccination after eradication of the disease[2].

Until 2022, MPXV outbreaks were typically confined to West and Central Africa, resulting from zoonotic spillover events with limited human-to-human transmission. Notable exceptions included a 2003 outbreak in the United States[3,4] and occasional exported cases[5]. However, in May 2022, an unprecedented global outbreak occurred,

affecting over 100,000 individuals across 122 countries[6,7]. This outbreak, driven by MPXV clade IIb, was characterized by sustained human-to-human transmission, primarily through sexual contact among men who have sex with men (MSM)[8]. The outbreak was eventually curtailed through a combination of effective risk communication, behavioral adaptation, and immunity acquisition through both infection and vaccination[9–11].

Despite this decline in case numbers, mpox remains a public health concern. A current outbreak linked to clade I in the Democratic Republic of the Congo, likely associated with a higher case fatality rate compared to clade IIb, highlights the urgent need for continued surveillance and improved diagnostics[12]. Long-term strategies must therefore prioritize the development of robust serological tools to support epidemiological monitoring and outbreak preparedness[13].

Serological assays capable of distinguishing infection-induced from vaccine-induced antibodies are critical to estimate true infection rates, detect unreported or asymptomatic cases, and evaluate the effectiveness of vaccination programs. Such tools also support studies of infection dynamics in at-risk populations[3]. However, MPXV-specific serology is challenging. Immunization with attenuated whole-virus vaccines induces polyclonal, cross-reactive antibody responses that closely resemble those generated by natural infection[14]. While this benefits vaccine-induced protection, it complicates serodiagnostic differentiation[15–17]. During the 2022 outbreak, large-scale vaccination campaigns adopted the modified vaccinia ankara (MVA) vaccine. The vaccine marketed as Imvanex, Jynneos, or Imvamune in Europe, the United States, and Canada, respectively, contains a highly attenuated virus originally developed as a safer alternative to the original vaccinia virus strains used for smallpox vaccination (e.g. Dryvax) and is now approved for MPXV immunization[18,19]. Notably, studies indicate that individuals vaccinated against smallpox in early childhood often retain detectable antibody levels throughout their lives, further aggravating the interpretation of antibody responses in MPXV-specific investigations[20,21].

In this study, we developed a machine learning (ML)-assisted bead-based serological multiplex assay to distinguish between MPXV infection, MVA vaccination, and orthopoxvirus-naïve status. IgG and IgM responses to 15 orthopoxvirus-specific antigens revealed distinct serological signatures, with ATI-N-CPXV emerging as a key discriminator alongside a cluster of consistently immunodominant antigens. To achieve robust classification, we trained six machine learning algorithms on two distinct serological cohorts. The serological cohorts comprised (i) acute-phase sera (acute cohort) and (ii) sera from an epidemiological study of an at-risk MSM population (epi cohort) and were tested both separately and combined. Among these algorithms, the Gradient Boosting Classifier (GBC) showed the highest performance and robustness. While optimal accuracy was achieved using all 15 antigens, a reduced panel of eight antigens (ATI-N-CPXV, ATI-C-CPXV, D8-VACV, E8-MPXV, A33-VACV, A35-MPXV, VACV lysate, and HEp-2 lysates, comprising Delta-VACV) displayed a strong performance, supporting assay consolidation. In an independent validation cohort, GBC ensemble predictions confirmed high classification accuracy for MPXV infections, including breakthrough cases, although distinguishing MVA-vaccinated from pre-immune or historically smallpox-vaccinated individuals remained challenging. Overall performance improved further when combining GBC with Linear Discriminant Analysis (LDA). Finally, removing the two most discriminative antigens only marginally reduced accuracy, demonstrating that redundancy among individually weaker antigens supports robust classification. Together, these findings establish ML-assisted multiplex serology as a scalable framework for orthopoxvirus surveillance, enabling accurate detection of MPXV exposure in complex immunological settings.

## Results

We developed an assay to differentiate MPXV-infected (MPXV group), MVA-vaccinated (MVA group), and unexposed individuals (pre-immune group) using 14 recombinant poxviral proteins known for serological reactivity, immunogenicity and/or inducing neutralizing antibodies (Fig. 1a, Table 1 and Supplementary Tables 1 to 8 for sequence homology)[22–30]. In addition to H3-VACV and A5-CPXV, we included five pairs of homologous recombinant proteins derived from VACV and MPXV. Additionally, as a potential marker for differentiation, we incorporated recombinant A-type inclusion (ATI) protein derived from CPXV, which has previously been used to distinguish between Dryvax and MVA vaccination[31]. To capture the complexity of the antibody response not covered by the recombinant proteins, we also incorporated lysates from VACV-infected and uninfected cells (included as a negative control). The antibody response to VACV-infected cell lysate with binding to uninfected cell lysate subtracted is hereafter referred to as Delta-VACV[17,32]. Finally, to complete the 19-plex antigen panel, we included three additional bead-bound controls: human serum albumin (HSA, to assess background binding) and anti-human IgG and anti-human IgM antibodies (as positive controls to confirm serum addition). This 19-plex assay was applied to all serological cohorts described in this study.

### Agreement with reference assays

We first evaluated the performance of our multiplex assay using serological cohorts previously characterized by established reference methods, including an enzyme-linked immunosorbent assay (ELISA), immunofluorescence assay (IFA), and neutralization test (NT)[16,17]. Agreement between our multiplex assay and these reference assays was high for both IgG and IgM measurements. Notably, we observed strong correlations between antibody binding to antigens in the multiplex assay and results from all three reference assays for both IgG and IgM detection (Supplementary Fig. 1). The strongest immune responses, characterized by a broad dynamic range (indicative of both high antibody affinity and abundance) were directed against Delta-VACV, D8-VACV/E8-MPXV, H3-VACV, A33-VACV/A35-MPXV, and B5-VACV/B6-MPXV (Supplementary Fig. 2). Other antigens tested also exhibited significant correlations with ELISA, IFA and NT results, albeit to a lesser extent. These findings confirmed that all tested orthopoxvirus-specific antigens were recognized by antibodies generated against viral infections, with certain antigens seeming to play a more dominant role in the overall immune response in our assay, and that our assay accurately reflects results obtained by well-established methods used to assess orthopoxvirus-specific serostatus.

### Serological cohorts used for assay development and testing

Next, we quantified both the IgG and IgM antibody immune response in larger cohorts of sera using our multiplex assay. To establish the assay and to train and test different ML-learning algorithms, we used two distinct serological cohorts (Fig. 1b and Table 2). The acute cohort included 307 sera from PCR-confirmed mpox cases (collected May–August 2022) in the acute and early convalescence phase (<33 days post-infection), as well as 37 sera collected two weeks post-MVA immunization (collected January 2020–October 2022). Pre-immune sera ($n = 193$) were obtained from pre-outbreak cohorts (2019) and/or from non-risk populations (suspected measles, mumps, rubella) to ensure no prior MPXV or MVA exposure. For distinguishing between infected, vaccinated, and non-exposed individuals, the acute cohort represents the best-case scenario, as samples were collected during the acute phase of infection or shortly after vaccination, while negative sera were from unexposed individuals.

The epi cohort (collected in April–June 2023) comprised sera from MSM attending sexually transmitted infections (STI) clinics. This population was at high risk of MPXV exposure and had a high rate of MVA vaccination. Classification of sera was based on self-reported

MPXV infection ($n = 59$), MVA vaccination ($n = 476$), or neither ($n = 324$), as indicated in a questionnaire completed by participants in a sero-epidemiological study[33]. Unlike the acute cohort, the epi cohort represents a more realistic dataset commonly encountered in sero-epidemiological studies, where samples are collected at later time points from at-risk populations with a higher likelihood of unrecog-nized infections. Working with these two cohorts allowed us to eval-uate the assay's performance under both controlled and real-world conditions. Importantly, the timing of serum collection relative to infection or vaccination varied between the two cohorts, enabling a broadly comparative analysis of immune responses at different post-exposure stages.

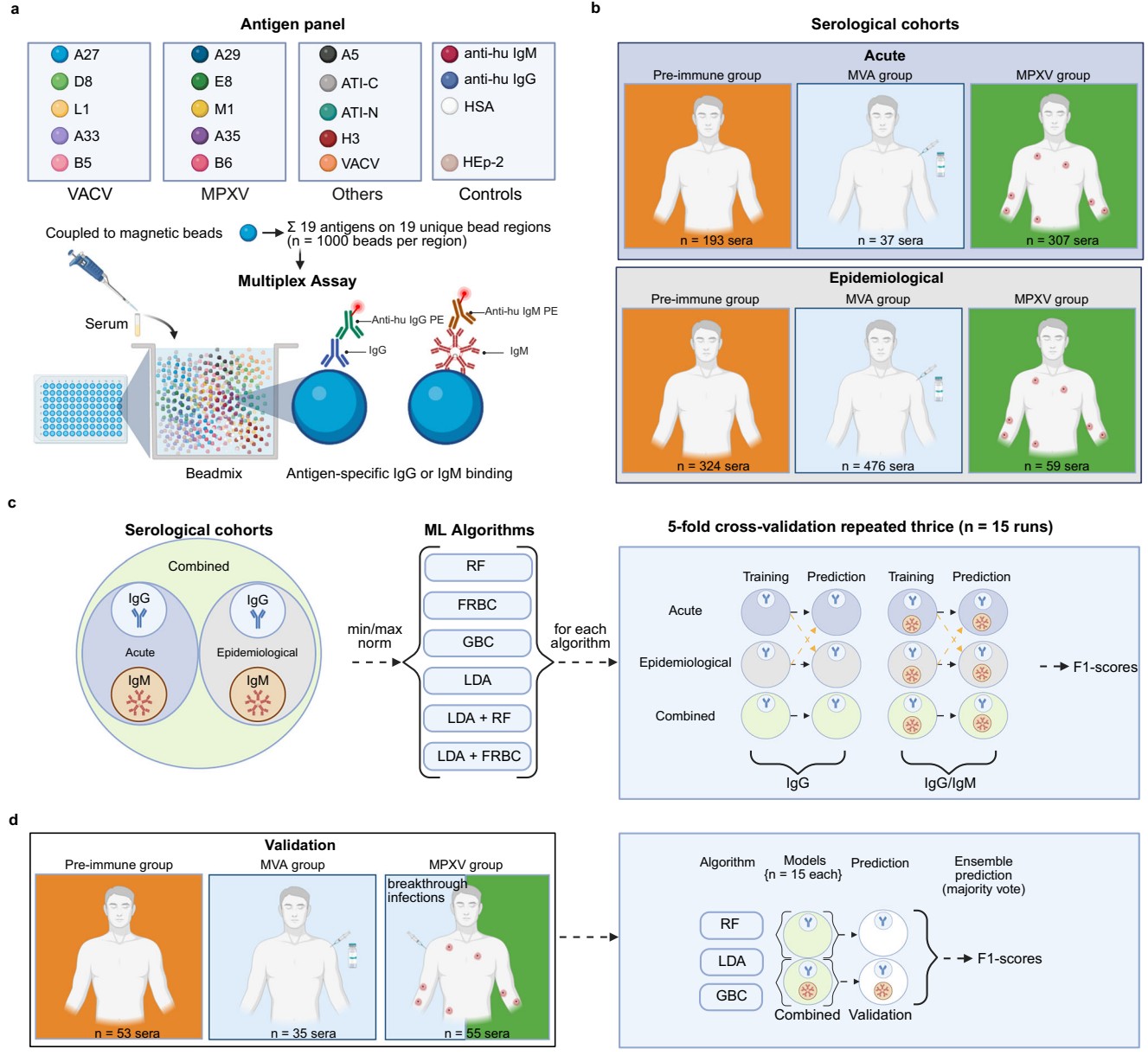

**Fig. 1 | Outline of antigen panel implemented in the multiplex assay, sample cohorts, and machine learning workflow for differentiating pre-immune, MVA-immunized, and MPXV-infected sera. a** Antigen panel and schematic repre-sentation of the bead-based multiplex immunoassay used to quantify antigen-specific IgG and IgM responses. This antigen panel was applied to all cohorts. **b** Serological cohorts included in the study to train the different ML models. The acute cohort comprises pre-immune sera (collected pre-pandemic), MVA sera (collected after MVA immunization), and MPXV sera (collected less than 33 days after confirmed infection). The epidemiological cohort consists of sera from a high-risk MSM population with self-reported MPXV infection and/or MVA immunization status. **c** Machine learning workflow. Normalized quantitative measurements of cohort samples were used to train six classifiers based on antigen-specific IgG or combined IgG/IgM responses: RF, FRBC, GBC, LDA, and two hybrid models com-bining LDA with RF or FRBC. Models trained on IgG data alone or combined

IgG/IgM data were evaluated cohort-specific using 5-fold cross-validation repeated three times (15 runs). Additionally, classifiers trained on the acute and epidemio-logical cohorts were cross-evaluated to assess dataset-specific performance. **d** Independent validation. A separate validation cohort was used to evaluate model generalizability. Final ensemble predictions were generated by majority voting across all 15 trained models per algorithm (RF, LDA, GBC) and antibody isotype, trained on the combined acute and epidemiological cohorts. Color coding: orange = pre-immune group, blue = MVA group, green = MPXV group. Abbrevia-tions: VACV, vaccinia virus; MPXV, monkeypox virus; MSM, men who have sex with men; HSA, human serum albumin; HEp-2, human epithelial type 2 cell extract; Anti-hu IgG/IgM PE: Anti-human IgG/IgM phycoerythrin (PE)-conjugated detection antibody; RF, Random Forest, FRBC, Fuzzy Rule-Based Classification, GBC, Gradient Boosting Classifier, LDA, Linear Discriminant Analysis; ML, machine learning. Cre-ated in BioRender. Stern, D. (2025) https://BioRender.com/kk4g6v5.

**Table 1 | Overview of poxviral antigens implemented in the multiplex assay**

| Name - source strain | Virion location | Remarks |
|---|---|---|
| A27-VACV /A29-MPXV | IMV membrane | Neutralizing mAbs[57,74,75] |
| L1-VACV /M1-MPXV | IMV membrane | Neutralizing mAbs[76] |
| D8-VACV /E8-MPXV | IMV membrane | Immunodominant, Neutralizing mAbs[26] |
| H3-VACV | IMV membrane | Immunodominant, Neutralizing mAbs[25] |
| A33-VACV / A35-MPXV | EEV membrane | EEV-neutralizing mAbs[23,28] |
| B5-VACV /B6-MPXV | EEV membrane | EEV-neutralizing mAbs[29] |
| A5-CPXV | Core protein | Immunodominant, cross reactive antigen[30] |
| ATI-C-CPXV/-N CPXV | A-type inclusion bodies | Differentiation of MVA/Dryvax[31] |
| Delta-VACV | Whole proteome | Normalized to uninfected cell lysate[17] |

*IMV* intracellular mature virion, *EEV* extracellular enveloped virion, *MPXV* monkeypox virus, *VACV* vaccinia virus, *CPXV* cowpox virus.

## Comparison of IgG and IgM patterns in different cohorts and age groups

We analyzed IgG and IgM responses across MPXV-infected, MVA-vaccinated, and pre-immune sera to look for distinguishing immune signatures. Additionally, we applied an age-based classification to assess the impact of presumed childhood smallpox vaccination as a potential confounder. Individuals older than 50 years as of 2022 (year of birth <1972) were classified as likely to have received smallpox vaccination in their early childhood due to mandatory vaccinations at that time, those under 40 as unlikely, and those between 40 and 50 as ambiguous and thus excluded from age-stratified analysis. Using data on year of birth (if available) allowed us to evaluate the impact of childhood vaccination on the antibody immune response and the differentiation between the three groups.

As shown in Fig. 2a, analysis of IgG responses in the acute cohort using a spider plot revealed a trend of increasing orthopoxvirus-specific antibody levels from pre-immune individuals to MVA-vaccinated individuals, with the highest responses observed in MPXV-infected individuals. In the pre-immune group, individuals with presumed childhood smallpox vaccination showed stronger IgG binding across all antigens, suggesting pre-existing immunity. Similarly, individuals with childhood smallpox vaccination had elevated IgG levels following both MVA vaccination and MPXV infection as compared to naïve subjects, indicative of booster effects in those previously exposed to smallpox vaccines. Comparing IgG responses between the acute and epi cohorts, we observed lower IgG levels post-MPXV infection in the epi cohort. Furthermore, immune response differences between the three serogroups were less pronounced in the epi cohort as compared to the acute cohort.

Analysis of IgM responses (Fig. 2b) shows that the highest IgM binding in the acute cohort was observed following MPXV infection, while IgM binding was largely absent in the epi cohort. Interestingly, in the epi cohort, a few sera in the pre-immune and MVA groups exhibited strong IgM responses, possibly indicating unrecognized infections.

## Antigen-specific contributions to the IgG response

We subsequently compared the immune responses to individual antigens across the three serogroups in each cohort (acute and epi). This analysis aimed to uncover potential differences that machine learning algorithms could leverage during training, thereby contributing to serogroup classification. Conversely, we also sought to identify potential confounding factors that could obscure or complicate the detection of genuine differences between serogroups.

Since IgM-mediated immune responses were substantial in the acute cohort but not in the epi cohort, we focused on IgG responses for comparison between the acute and epi cohorts. As shown in Fig. 2c (acute) and d (epi), we identified D8-VACV/E8-MPXV, Delta-VACV, B5-VACV/B6-MPXV, and A33-VACV/A35-MPXV as the most significant

contributors to the overall IgG response (E8-/B6-/A35-MPXV shown in Fig. 2c, d, Supplementary Fig. 3 for other antigens, Supplementary Fig. 4 for IgM results). A significant difference between MPXV-infected and pre-immune sera across both cohorts was observed for every antigen. However, due to substantial overlap between the MVA and MPXV serogroups, these antigens alone were insufficient to reliably differentiate MPXV infection from MVA immunization. In contrast, ATI-N-CPXV demonstrated a highly distinctive immune profile. Highly significantly elevated IgG levels against ATI-N-CPXV were found only in MPXV-infected individuals, distinguishing them from both pre-immune and MVA-immunized individuals, regardless of childhood smallpox vaccination status and cohort, although the difference was less pronounced in the epi cohort.

Finally, we evaluated whether differential binding to homologous VACV and MPXV antigen pairs could contribute to differentiation between the three serostatuses. To this end, we calculated the ratios of MPXV-to-VACV antigen binding and tested whether these ratios significantly differed among pre-immune, MVA, and MPXV-infected groups (Fig. 2e, f, Supplementary Fig. 5). We compared the IgG binding responses to MPXV- versus VACV-derived proteins for the immunodominant antigen pairs tested and observed a stronger binding to MPXV proteins in the MPXV group and to VACV proteins in the MVA and pre-immune group for the following antigen pairs: A33-VACV/A35-MPXV, B5-VACV/B6-MPXV, D8-VACV/E8-MPXV. The effect was more pronounced in younger individuals (aged 40 and under).

Taken together, these antigen-specific patterns laid the foundation for our machine learning classification strategies.

## Classification performance of ML-guided analysis

We next trained supervised ML models to classify sera as pre-immune, MVA-vaccinated, or MPXV-infected (Fig. 1c). Although some immune responses were likely influenced by childhood smallpox vaccination, we did not stratify by age due to incomplete metadata across the cohorts. Six classifiers were tested: Random Forest (RF) as a baseline model, Gradient Boosting Classifier (GBC), Linear Discriminant Analysis (LDA), Fuzzy Rule-Based Classification (FRBC), and two hybrid models (LDA + RF, LDA + FRBC) using LDA for dimensionality reduction followed by classification by RF or FRBC. A deep neural network (DeepTables) was also evaluated but excluded from primary analysis due to poor performance, likely reflecting the dataset's small size, tabular structure, and lack of extensive hyperparameter tuning (see Supporting Data[34]).

All orthopoxvirus antigens were used for training except L1-VACV and M1-MPXV, which showed inconsistent coupling (Supplementary Fig. 6). Additionally, 132 sera from the epi cohort were excluded from training due to suspected unrecognized MPXV exposure, based on unexpected orthopoxvirus reactivity or ATI-N-CPXV binding in self-reported unexposed individuals under 40 years (Supplementary Figs. 7 and 8).

**Table 2 | Collection date of MPXV positive sera, age distribution, and childhood vaccination status of subjects in the acute and epidemiological serological cohorts**

| Serostatus | Age group | Childhood Immunization | Acute cohort May to August 2022 Percent (Serostatus) | n | Epidemiological cohort April to June 2023 Percent (Serostatus) | n |
|---|---|---|---|---|---|---|
| MPXV | <30 | No | 9.5% | 29 | 20.3% | 12 |
| | <40 | No | 43.3% | 132 | 49.2% | 29 |
| | <50 | Ambiguous | 33.6% | 103 | 18.6% | 11 |
| | <60 | Yes | 9.8% | 30 | 8.5% | 5 |
| | 60+ | Yes | 3.6% | 11 | 3.4% | 2 |
| | NA | – | 0.7% | 2 | – | – |
| | Sum | – | – | 307 | – | 59 |
| MVA | <30 | No | 0.0% | 0 | 20.6% | 98 |
| | <40 | No | 43.2% | 16 | 40.5% | 193 |
| | <50 | Ambiguous | 29.7% | 11 | 23.5% | 112 |
| | <60 | Yes | 27.0% | 10 | 12.4% | 59 |
| | 60+ | Yes | 0.0% | 0 | 2.9% | 14 |
| | NA | – | – | – | – | – |
| | Sum | – | – | 37 | – | 476 |
| Pre | <30 | No | 26.9% | 28 | 36.7% | 119 |
| | <40 | No | 15.4% | 16 | 31.8% | 103 |
| | <50 | Ambiguous | 14.4% | 15 | 13.9% | 45 |
| | <60 | Yes | 14.4% | 15 | 12.0% | 39 |
| | 60+ | Yes | 28.8% | 30 | 5.6% | 18 |
| | NA | – | – | 89 | – | – |
| | Sum | – | – | 193 | – | 324 |
| All | <30 | No | 12.8% | 57 | 26.7% | 229 |
| | <40 | No | 36.7% | 164 | 37.8% | 325 |
| | <50 | Ambiguous | 28.9% | 129 | 19.6% | 168 |
| | <60 | Yes | 12.3% | 55 | 12.0% | 103 |
| | 60+ | Yes | 9.2% | 41 | 4.0% | 34 |
| | NA | – | – | 91 | | – |
| | Sum | – | – | 537 | | 859 |

*MPXV* monkeypox virus, *MVA* Modified Vaccinia Ankara, *Pre* pre-immune, *NA* not available.

We trained the models using either normalized (min-max scaling) IgG data alone or combined normalized IgG/IgM data to determine whether the inclusion of IgM would improve classification performance. Models were tested on both matched (the same serum cohort was used for model training and testing) and mismatched (different serum cohorts were used for model training and testing) cohorts through 5-fold cross-validation, repeated thrice. For mismatched (different serum cohorts were used for model training and testing) evaluation, models were trained on the entire source cohort and tested on the entire target cohort. Additionally, we trained all models on a combined dataset containing results from both cohorts, acute and epi.

When comparing F1 scores across test datasets, we observed notable differences between the tested cohorts and the applied ML algorithms (Fig. 3a and Supplementary Table 9). Overall, as shown for the algorithms trained on IgG/IgM results from both cohorts combined, GBC achieved the overall highest performance (mean F1 ± SD: 0.82 ± 0.02; calculated from macro-averaged F1 scores from n = 15 runs from the cross-validation), indicating strong discriminative power, followed by LDA (0.80 ± 0.02) and LDA + RF (0.80 ± 0.02). RF (0.76 ± 0.03) and FRBC (0.75 ± 0.02) performed slightly worse, with LDA + FRBC showing the weakest performance (0.60 ± 0.04), suggesting that FRBC may have been more affected by the LDA-based dimensionality reduction than RF. Compared to the

baseline random forest model, the GBC algorithm achieved an F1 score that was 0.06 higher, representing a meaningful improvement and demonstrating the superiority of GBC for this classification task. The highest performance was achieved when models were trained and tested on the same cohort. When training and testing across mismatched cohorts, model performance dropped substantially (mean F1 < 0.6). Importantly, combining data from both cohorts (combined) restored model performance across algorithms, indicating increased generalizability.

To quantify the contribution of individual antigens to model performance, we performed recursive feature elimination on the GBC model (Supplementary Fig. 9). By including all antigens the highest performance was achieved (baseline F1 = 0.82 ± 0.02 for IgG/IgM data), although five antigens could be removed with only a minor reduction in accuracy (F1 = 0.81 ± 0.02). Performance began to decline when ATI-C-CPXV and A33-VACV were removed, and even more when Delta-VACV (F1 = 0.77 ± 0.01) was removed. The largest drops occurred when A35-MPXV, ATI-N-CPXV, and D8-VACV were removed, leaving E8-MPXV as the strongest single antigen (F1 = 0.50 ± 0.02 for IgG; 0.62 ± 0.03 for combined IgM/IgG). These analyses indicate that a reduced panel of eight antigens (ATI-N-CPXV, ATI-C-CPXV, D8-VACV, E8-MPXV, A33-VACV, A35-MPXV, VACV lysate, and HEp-2 lysates, comprising Delta-VACV) achieves performance comparable to the full panel.

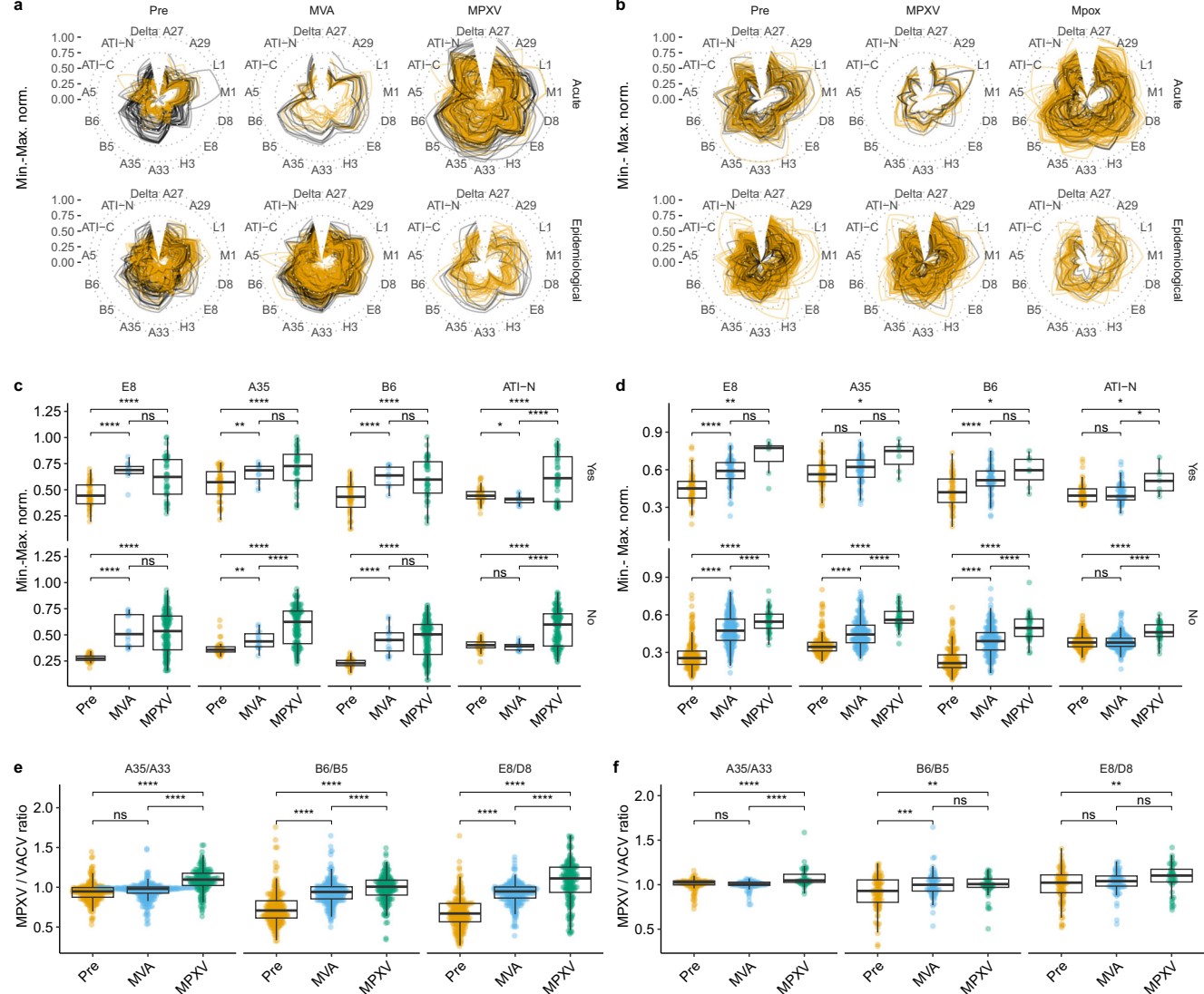

**Fig. 2 | Antibody profiles of serological cohorts measured by multiplex assay for IgG and IgM binding. a, b** Spider plots showing normalized IgG (**a**) and IgM (**b**) responses (min−max scaled between 0 and 1) to each antigen in the acute (top) and epi (bottom) cohorts, stratified by serostatus group (Pre, MVA, MPXV). Responses were further stratified by presumed childhood smallpox vaccination status: black lines indicate vaccinated individuals (age ≥50 years), orange lines unvaccinated individuals (age < 40 years). **c, d** Box plots of min−max normalized IgG responses to selected antigens in the acute (**c**) and epi (**d**) cohorts in each serostatus group (Pre, MVA, and MPXV). Upper panels (Yes) show presumed childhood smallpox vaccinated individuals, lower panel (No) sera from unvaccinated individuals regarding childhood smallpox vaccination. **e, f** Ratios (dimensionless) of IgG binding to homologous MPXV and VACV antigen pairs (A35/A33, B6/B5, E8/D8) in the combined acute and epi cohorts, stratified by serostatus groups and presumed smallpox vaccination status (**e**: naïve; **f**: childhood smallpox vaccinated). Statistical

significance between serogroups (two-sided t test; no adjustments were made for group sizes of vaccinated and unvaccinated individuals: Acute cohort: Pre: 45 and 44; MVA: 10 and 16; MPXV: 41 and 161. Epi cohort: Pre: 57 and 222; MVA: 73 and 291; MPXV: 7 and 41. Boxes represent the interquartile range (IQR; 25th–75th percentile), with the line inside indicating the median. Whiskers extend to the most extreme data points within 1.5 × IQR from the hinges. Individual data points are shown using the beeswarm package in R. Color coding (**c**–**f**): orange = pre-immune group, blue = MVA group, green = MPXV group. Abbreviations: epi, epidemiological; AU, arbitrary units; Pre, pre-immune; VACV, vaccinia virus; MPXV, monkeypox virus; MVA, Modified Vaccinia Ankara; Min.-Max. norm., indicates min−max normalized values scaled between 0 and 1 (unitless). Source data are provided as a Source Data file. Statistically significant differences between serogroups (t test, two-sided) indicated as asterisks (ns; *$p < 0.05$; **$p < 0.01$; ***$p < 0.001$; ****$p < 0.0001$). Exact $p$-values are provided in the Source Data file.

## Specificity of ML-based predictions after childhood vaccination

Based on overall performance, we focused further analyses on GBC and LDA, the two most robust classifiers trained on the combined dataset. For comparison, and to serve as a strong baseline due to its stability and interpretability, we included the performance metrics achieved by the RF model. To assess the potential confounding effect of childhood smallpox vaccination, we stratified misclassification rates by age as a proxy for vaccination status, as we explained earlier.

As shown in Fig. 3b, among presumed smallpox vaccinated individuals, ~30% of pre-immune sera were misclassified as MVA by both

GBC and LDA, but 65% by RF, while 14% (GBC), 7% (LDA), and 12% (RF) were misclassified as MPXV. In contrast, younger, presumed non-vaccinated individuals showed lower misclassification rates of ~2%. These findings suggest that age-associated residual immunity contributes to orthopoxvirus-specific antibody profiles, complicating accurate classification especially for RF.

As shown in Fig. 3c, confusion matrices confirmed high classification accuracy for both models. Between LDA and GBC, LDA produced fewer false positives among pre-immune samples (LDA: 48; GBC: 64) while GBC less often classified

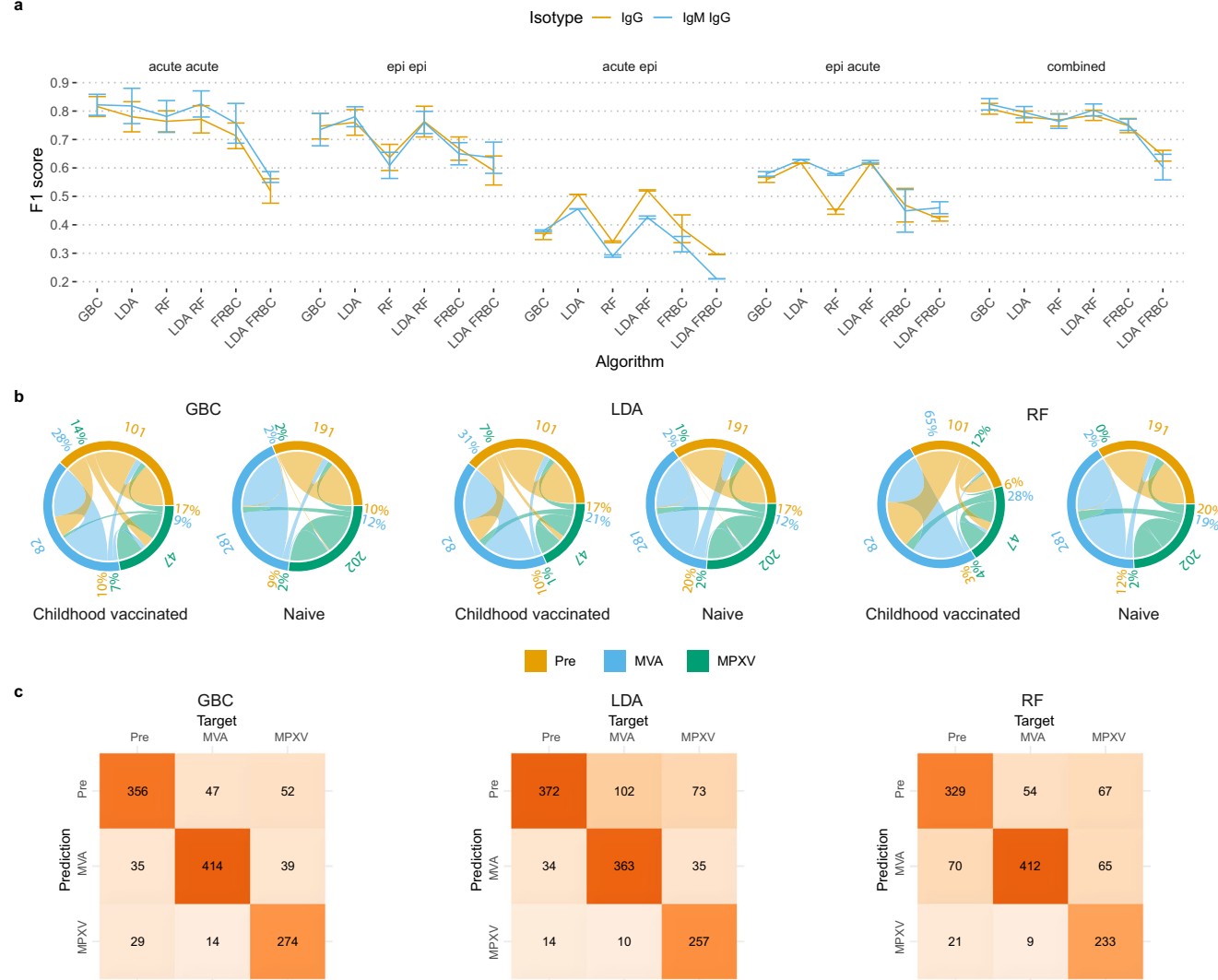

**Fig. 3 | Performance of machine learning-based serogroup classification.**
**a** Classification performance (F1 score; mean ± SD from 15 cross-validation runs) across different combinations of training and test datasets (acute, epi, and combined) using a 5-fold cross-validation repeated three times (*n* = 15). Results are shown for each machine learning algorithm tested, stratified by input type (IgG only vs. IgG/IgM). **b** Circular plots showing classification outcomes of the combined cohort using GBC, LDA, and RF on IgG/IgM datasets. Outer segments represent the true serogroup (number of sera shown), while inner traces indicate model predictions (misclassifications shown as percentages, colored by prediction of

misclassification), further stratified by presumed smallpox vaccination status or presumed naïve individuals. **c** Confusion matrices for ensemble predictions using GBC, LDA, and RF on the combined serological cohort and IgG/IgM datasets used for establishment of the ML algorithms. Abbreviations: epi, epidemiological; pre, pre-immune; VACV, vaccinia virus; MPXV, monkeypox virus; MVA, Modified Vaccinia Ankara; LDA, Linear Discriminant Analysis; GBC, Gradient Boosting Classifier; RF, Random Forest; FRBC, Fuzzy Rule-Based Classification; ML, machine learning. Source data are provided as a Source Data file.

MPXV-infected and MVA-vaccinated individuals as pre-immune (LDA: 175; GBC: 99).

## Validation of the assay performance

To further assess model generalizability, we validated RF, GBC, and LDA classifiers on an independent validation cohort (*n* = 143) distinct from the training cohorts (Fig. 1d). This independent validation cohort included 55 MPXV-infected sera, 32 of which were breakthrough infections following one (*n* = 13) or two (*n* = 19) MVA vaccine doses, 35 post-MVA vaccination sera, and 53 pre-immune sera, including 23 individuals born before 1972 who likely had pre-existing smallpox immunity. For classification, we applied an ensemble learning approach based on the majority vote across all 30 predictions from the IgG and IgG/IgM models generated during cross-validation on the combined serological cohorts. To quantify classification uncertainty, we computed mean macro

F1 scores and corresponding 95% confidence intervals using bootstrap analysis with 2000 iterations.

Classification results are visualized alongside a heatmap of scaled IgG responses, annotated with serostatus relative to the Delta-VACV antigen (positive or negative), the number of recorded MVA vaccinations (where available), and classification outcomes from LDA, RF, and GBC using the ensemble learning algorithm (Fig. 4a, Supplementary Table 10). MVA-vaccinated sera were mostly correctly classified (32–34 of 35 sera, > 90%). In the pre-immune group, misclassifications were more frequent (23–32 of 53 sera; 43–60%), particularly among older (year of birth <1972) individuals (Supplementary Fig. 10). Most of these misclassified samples (~70–90%) were reassigned to the MVA group, consistent with residual immunity from childhood smallpox vaccination. Among MPXV cases, GBC correctly classified most samples (80%), including breakthrough infections, while LDA more frequently misclassified these as MVA (~45% of MPXV misclassified as MVA).

The overall performance of all ML models decreased on the independent validation cohort compared to the combined cohort (Table 3, mean F1 score for GBC of 0.70, 95% CI: 0.63–0.78). On the independent validation cohort, GBC improved macro-F1 by +0.11 vs LDA and +0.02 vs RF; on the combined cohort, gains were +0.04 vs LDA and +0.06 vs RF. As misclassifications more frequently occurred between MVA and childhood vaccinated pre-immune sera but not between MPXV and MVA or pre-immune sera, the per-class sensitivity and specificity for correctly classifying MPXV infections was high in the independent validation cohort, with a sensitivity of 0.92 (95% CI, 0.83–0.98) and specificity of 0.88 (95% CI, 0.82–0.94), similar to what was observed on the combined cohort (sensitivity: 0.86, 95% CI 0.83–0.90; specificity: 0.90, 95% CI: 0.88–0.92), indicating robust detection capability for MPXV infections (Supplementary Table 11).

Finally, we evaluated an ensemble approach combining two algorithms to enhance classification (Table 3). Motivated by LDA's higher accuracy for pre-immune sera and GBC's superior detection of MPXV infections (Fig. 3c), we implemented a hybrid ensemble-learning approach (Fig. 4c) that applies LDA to seronegative samples (defined by Delta-VACV binding) and GBC to seropositive samples. In the combined cohort, the ensemble performed comparably to LDA and slightly below GBC; on the independent validation cohort it slightly improved performance to F1 = 0.76 (95% CI, 0.68–0.83), although the difference was not significant due to largely overlapping confidence intervals. Finally, we incorporated prediction confidence as an added QC metric. By retaining only predictions above a predefined threshold (>0.5 LDA posterior and ensemble probability), users can focus on high-confidence outputs, increasing trust and potentially improving performance. Indeed, including only sera above the threshold slightly increased the F1 score to 0.85 (95% CI, 0.83–0.88) for GBC in the combined cohort (n = 1,156 sera retained) and to 0.77 (95% CI, 0.69–0.84; GBC) and 0.83 (95% CI, 0.76–0.90; ensemble serostatus; n = 124 of 143 sera retained) in the independent validation cohort. However, neither the hybrid ensemble strategies, nor excluding low confidence predictions significantly outperformed GBC alone across the combined cohort, underscoring the strength of GBC as a standalone classifier.

## Comparison with single-antigen classifiers

Previous studies used classifiers based on single or few antigens to distinguish MPXV infection from other sera or to identify orthopoxvirus seropositivity resulting from infection or vaccination[35,36]. To directly quantify the performance gain achieved by our ML-based approach, we systematically compared our GBC model (baseline, without classification enhancement) against simplified single-, dual-, or even triple-antigen classifiers on the same cohorts. This head-to-head evaluation was essential, as comparisons based solely on published performance metrics can be misleading due to differences in cohort composition, sample size, and sampling timepoints.

Using ROC analyses to determine threshold values and to compare the performance of single antigens for distinguishing seropositive from seronegative samples, E8-MPXV emerged as the best single antigen, followed by B6-MPXV, whereas ATI-N-CPXV and the A35-MPXV/A33-VACV binding ratio best differentiated MPXV-infected from MVA-vaccinated samples (Supplementary Fig. 11, Quantitative cut-offs in log10-ng/mL are in Supplementary Table 12). For multiclass classification, we evaluated rule-based schemes using pairs of top-performing antigens to approximate the ML classes (pre-immune, MVA, MPXV): (i) ATI-N-CPXV + E8-MPXV and (ii) the A35-MPXV/A33-VACV binding ratio + B6-MPXV. Using ROC-derived thresholds, samples were assigned as follows: MPXV if ATI-N-CPXV positive (or ratio positive); MVA if seropositive (E8-MPXV or B6-MPXV positive) but ATI-N/ratio negative; pre-immune if both markers were negative.

Across all tasks, the GBC-based ML approach consistently outperformed simplified classifiers (Supplementary Table 13). While

binary serostatus classification using single antigens approached ML performance (F1 score decrease of only 0.02–0.05), classification of MPXV-infected sera and multiclass predictions showed markedly poorer performance (F1 score decrease of 0.12–0.20). Importantly, reliance on second-tier antigens, such as the A35-MPXV/A33-VACV ratio, resulted in a substantial performance drop on the validation panel (F1 score decrease of 0.20), underscoring the unique value of ATI-N-CPXV as a single discriminator of MPXV-infections.

In contrast, our GBC model demonstrated remarkable robustness. Removing ATI-N-CPXV, E8-MPXV, or D8-VACV individually, or even both ATI-N-CPXV and E8-MPXV, had a negligible effect on the classification performance (F1 decrease ≤ 0.01–0.02 across combined and validation cohorts; Supplementary Table 14). Even the removal of all three antigens reduced performance only modestly (F1 decrease = 0.04 on the combined cohort; F1 decrease = 0.01 on validation). Thus, our ML-based approach not only outperforms simpler single-, dual, or triple-antigen strategies but also provides greater robustness and redundancy, enabling stable classification even when key antigenic markers are removed.

## Discussion

Accurate serological discrimination between MPXV infection and MVA vaccination is challenging due to extensive antigenic cross-reactivity. In this study, we demonstrate that combining a high-throughput multiplex assay with machine learning enables accurate classification of serum samples as pre-immune, MVA-vaccinated, or MPXV-infected —even in complex immunological landscapes shaped by prior vaccination or breakthrough infections.

Conventional serological assays struggle to differentiate immune responses between closely related orthopoxviruses due to high sequence homology among immunodominant antigens[17,32,37,38]. Prior approaches, such as pre-absorption techniques[39–41], differential antigen binding assays using heterologous antigens[14,42], or MPXV-specific short peptides or proteins[36,43–45], have provided promising results regarding the specificity and sensitivity, but remain either technically limited or should be further validated. Additionally, previously reported sensitivities and specificities in the literature vary depending on the assay and setup and may range between 93%/98% (sensitivity/ specificity) for a combination of mpox-specific peptides and recombinant proteins[45], 86%/90% for an ELISA with MPXV-specific peptides[46], 86%/100% for a post-absorption ELISA[39], 100%/90% for another peptide-specific ELISA and 92-100%/100% for a combination between titre to VACV/MPXV binding and titre decline[47]. Finally, to differentiate MPXV-infected subjects from MVA-vaccinated, recently published multiplex assays similar to our approach achieved sensitivities/specificities of 88%/97% and 89%/80%[35,36]. However, confounding factors such as differences in antigen selection, sample size, time between infection and sampling, and the composition of negative cohorts (e.g., vaccination history, population risk) can substantially influence assay performance, rendering direct comparisons potentially misleading.

Despite these caveats our assay compares favorably with prior work, yielding 86–94% sensitivity and 88–93% specificity for distinguishing MPXV infection from both pre-immune and MVA-vaccinated sera, depending on the cohort and whether low-confidence predictions are excluded. Crucially, the best-performing model (GBC) consistently outperformed single-, dual-, and triple-antigen rule-based classifiers across binary and multiclass tasks in both the combined dataset and the independent validation cohort. These findings demonstrate a clear performance advantage of ML-based multiplex serology over simplified approaches.

Furthermore, compared to previously published studies, our work is distinguished by the large number of sera analyzed, the inclusion of a comprehensive panel of 15 unique antigens, the measurement of both IgG and IgM responses, and evaluation of both acute and convalescent phase sera. Moreover, our results are validated using

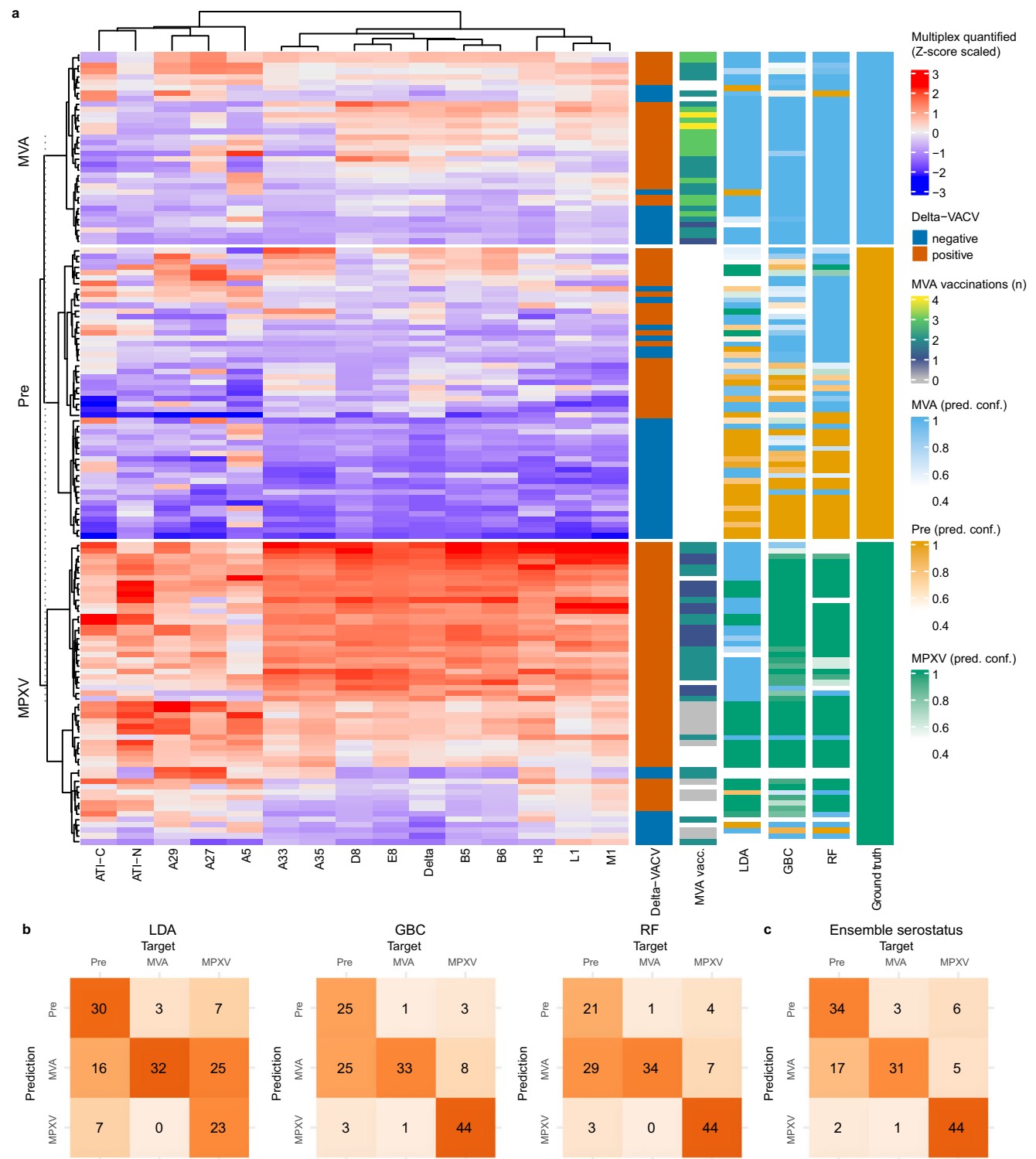

**Fig. 4 | Machine learning-based classification of orthopoxvirus serogroups in an independent validation cohort. a** Heatmap of Z-score scaled (centered and standardized per antigen) IgG multiplex measurements ($n = 143$), hierarchically clustered using the Ward D2 algorithm. Associated annotations show: IgG-based serostatus inferred from Delta-VACV binding; reported number of MVA vaccine doses; classifications for each serogroup category (Pre, MVA, MPXV) across 15 models (RF, LDA, and GBC, trained on IgG or IgG+IgM); ground truth serogroup labels. Prediction confidence (pred. conf.) is indicated by color shading. **b** Confusion matrices showing classification results from LDA, RF, and GBC. **c** Confusion matrix for ensemble predictions from LDA and GBC, stratified by Delta-VACV serostatus (LDA applied to seronegative, GBC to seropositive). Color coding: orange = pre-immune group, blue = MVA group, green = MPXV group. Abbreviations: Pre, pre-immune; VACV, vaccinia virus; MPXV, monkeypox virus; MVA, Modified Vaccinia Ankara; LDA, Linear Discriminant Analysis; GBC, Gradient Boosting Classifier; RF, Random Forest.

**Table 3 | Performance parameters (mean and 95% CI, calculated by bootstrap analysis with 2000 iterations) of ensemble learning algorithms on the combined serological cohort and independent validation cohort**

| Cohort | Algorithm[a] | Filtered[b] | n | F1 | Precision | Recall |
|---|---|---|---|---|---|---|
| Combined | LDA | None | 1260 | 0.79 (0.77–0.81) | 0.81 (0.79–0.83) | 0.78 (0.76–0.81) |
| Combined | RF | None | 1260 | 0.77 (0.74–0.79) | 0.79 (0.77–0.81) | 0.76 (0.74–0.79) |
| Combined | GBC | None | 1260 | 0.83 (0.80–0.85) | 0.83 (0.81–0.85) | 0.82 (0.80–0.84) |
| Combined | Ensemble serostatus | None | 1260 | 0.80 (0.78–0.82) | 0.82 (0.79–0.84) | 0.80 (0.78–0.82) |
| Independent Validation | LDA | None | 143 | 0.59 (0.51–0.67) | 0.65 (0.58–0.73) | 0.63 (0.56–0.70) |
| Independent Validation | RF | None | 143 | 0.68 (0.61–0.76) | 0.74 (0.67–0.81) | 0.72 (0.66–0.78) |
| Independent Validation | GBC | None | 143 | 0.70 (0.63–0.78) | 0.76 (0.70–0.83) | 0.74 (0.67–0.80) |
| Independent Validation | Ensemble serostatus | None | 143 | 0.76 (0.68–0.83) | 0.77 (0.70–0.84) | 0.78 (0.70–0.84) |
| Combined | LDA | Confidence > 0.5 | 1140 | 0.83 (0.80–0.85) | 0.85 (0.83–0.86) | 0.82 (0.80–0.84) |
| Combined | RF | Confidence > 0.5 | 1173 | 0.80 (0.78–0.82) | 0.82 (0.80–0.84) | 0.79 (0.77–0.82) |
| Combined | GBC | Confidence > 0.5 | 1156 | 0.85 (0.83–0.88) | 0.86 (0.84–0.88) | 0.85 (0.83–0.87) |
| Combined | Ensemble serostatus | Confidence > 0.5 | 1155 | 0.83 (0.81–0.85) | 0.84 (0.83–0.86) | 0.83 (0.81–0.85) |
| Independent Validation | LDA | Confidence > 0.5 | 124 | 0.65 (0.57–0.73) | 0.74 (0.66–0.81) | 0.68 (0.61–0.75) |
| Independent Validation | RF | Confidence > 0.5 | 124 | 0.75 (0.67–0.82) | 0.81 (0.74–0.87) | 0.78 (0.72–0.84) |
| Independent Validation | GBC | Confidence > 0.5 | 124 | 0.77 (0.69–0.84) | 0.81 (0.75–0.87) | 0.79 (0.72–0.85) |
| Independent Validation | Ensemble serostatus | Confidence > 0.5 | 124 | 0.83 (0.76–0.90) | 0.85 (0.79–0.91) | 0.84 (0.78–0.91) |

Number of sera used for calculating the performance parameters.

*LDA* linear discriminant analysis, *RF* random forest, *GBC* gradient boosting classifier.

[a]Algorithm used for serostatus prediction.

[b]Filtered = probability > 0.5 (ensemble probability; LDA posterior).

an independent cohort and include samples from individuals with a history of smallpox vaccination. A significant aspect of our work is the application of machine learning algorithms to multiplex serological data. Although the extensive data generated by multiplex assays make machine learning an intuitive analytical choice, few studies have applied such approaches for differentiating between various serogroups. Previous studies applied machine learning to serological multiplex data to distinguish recent from historical malaria transmission[48,49], to improve serodiagnosis for SARS-CoV-2 (Severe Acute Respiratory Syndrome Coronavirus 2)[50] or to identify cases of active tuberculosis[51].

There are several key features of our work that highlight the strengths of combining the multiplex assay with ML-based analysis.

First, by including a panel of highly immunogenic orthopoxvirus proteins, our assay closely mirrors the results of established reference methods while also capturing the complex immune response against orthopoxviruses. In our study several highly reactive antigens including H3-VACV, E8-MPXV/D8-VACV, A35-MPXV/A33-VACV, and B6-MPXV/B5-VACV proved to be excellent markers for differentiating seropositive from seronegative samples[32,52–56], while others such as M1-MPXV/L1-VACV, despite still being recognized, exhibited lower reactivity and discriminatory power in our assay[57]. Our data indicate that differentiation between pre-immune and MVA/MPXV groups primarily depends on overall signal strength from highly reactive antigens. Distinction between MVA and MPXV is predominantly driven by reactivity against ATI-N-CPXV, while differential binding to MPXV and VACV homologous antigens also contributes[14]. This aligns with prior work identifying MPXV protein A27—the homolog of ATI-N-CPXV—as a key discriminator of infection[36], as well as binding to A35-MPXV/A33-VACV and B6-MPXV/B5-VACV homologous antigen pairs[14,35]. ATI has also been used to distinguish Dryvax from MVA immunization, as MVA-induced responses to ATI are absent due to a genomic truncation[31]. Misclassification of older individuals as MVA-vaccinated suggests that ATI-specific immunity from historic small-pox vaccination has largely waned, while responses to other ortho-poxvirus antigens remain detectable for decades[20]. This enables discrimination between MPXV infection, MVA vaccination, and

residual smallpox immunity yet also indicates that ATI-N-CPXV may be more informative in the early post-infection period, with decreasing sensitivity over time.

Across binary and multiclass tasks, ML-based classification consistently outperformed single-, dual-, and triple-antigen rules evaluated on the same sera. Even when restricted to the strongest single markers, ML, particularly GBC, achieved higher accuracy for identifying MPXV infections, which is the endpoint most relevant for seroepidemiology. These gains indicate greater robustness to confounders (e.g., residual smallpox immunity) and improved transferability to independent cohorts compared with rule-based approaches. Notably, our benchmarking demonstrates that the performance of the top-performing ML algorithm (GBC) was largely unaffected when ATI-N-CPXV or even ATI-N-CPXV together with E8-MPXV, the two most influential antigens, were excluded from the training and testing dataset. This finding indicates that the remaining antigens provide sufficient redundancy to capture key patterns in the antibody response, enabling reliable discrimination between MPXV-infected, MVA-immunized, and pre-immune sera even in the absence of the strongest antigens. These results underscore the advantages of ML-based multiplex serology over classical single-antigen approaches, highlighting not only the robustness of the ML method, but also its ability to maintain high classification performance even when major antigenic markers are missing.

Second, by evaluating several different ML algorithms across epidemiologically diverse serological cohorts, we emphasize the importance of robust training data. As our data showed, training and testing ML models on mismatched serological cohorts (acute vs. epidemiological) reduced prediction accuracy, highlighting the importance of dataset selection to account for factors such as timing of infection, prior MVA immunization, and waning antibody responses. This diversity is essential for developing ML models with strong generalizability and reliable performance.

Third, validation on an independent dataset confirmed the robustness of our approach. The validation cohort included challenging cases like breakthrough MPXV infections and people with immunity from childhood vaccinations. While older individuals were

more often misclassified, our ML-based assay, especially the GBC model, still showed strong predictive performance.

Our comparative analysis of LDA and GBC highlights key algorithmic trade-offs relevant for serological classification. The lower performance of LDA in distinguishing MPXV breakthrough infections from MVA likely stems from its assumptions of Gaussian-distributed features and a shared covariance matrix, which may not accurately reflect the antigen distribution (Supplementary Fig. 12). If antigen responses are non-normally distributed or exhibit complex interactions, LDA's linear boundaries struggle to separate the classes. Such assumptions are unlikely to hold in serological datasets, which frequently exhibit skewed or bimodal distributions driven by factors such as vaccination history, breakthrough infections, and waning antibody responses. These distributional characteristics likely contributed to the reduced performance of LDA, particularly in distinguishing complex or overlapping immune states. In contrast, GBC can model nonlinear relationships and is more robust to outliers, making it better suited for capturing subtle immune response patterns that LDA may overlook[58]. Notably, GBC achieved this level of performance without explicit modeling of prior MVA vaccination before MPXV infection or childhood smallpox immunization during training. When evaluated on an independent dataset, GBC alone demonstrated strong predictive power in the detection of breakthrough infections.

Despite these promising results, several limitations remain. The inherent variability in humoral immune responses can lead to misclassification, particularly among individuals with weak or waning antibody levels. Childhood smallpox immunization emerged as a relevant confounder, with many such individuals misclassified as MVA-vaccinated, which is a plausible outcome given that both exposures elicit similar immune profiles. As metadata for age was incomplete, we could not stratify for this important confounder. From a seroepidemiological perspective, it is reassuring that misclassification as MPXV-infected was observed far less frequently. Notably, the ML models provided confidence scores for each prediction, enabling the identification of uncertain classifications. Nevertheless, further optimization could improve performance in borderline cases.

Future work could consider incorporating more antigens with higher specificity and explicitly stratifying models by breakthrough infection status and childhood immunization history. Explicitly considering HIV status in future studies would enable investigation of the humoral immune response in people with HIV. This population is at increased risk for mpox infection and may experience different clinical outcomes or reduced effectiveness of MVA vaccination[59,60]. However, vaccination significantly reduces the severity of symptoms in breakthrough infections. Conversely, we demonstrate that high classification performance of the GBC algorithm can be maintained with a reduced panel of just eight antigens, namely ATI-N-CPXV, ATI-C-CPXV, D8-VACV, E8-MPXV, A33-VACV, A35-MPXV, VACV lysate, and HEp-2 lysates, comprising Delta-VACV. Selecting the most informative antigens enables minimization of multiplex assay costs while preserving, and potentially optimizing, overall assay performance.

Advances in assay technology, such as simultaneous IgG and IgM detection using the latest Luminex platforms[61], may further streamline and enhance the assay's diagnostic utility. Our results demonstrate that including IgM measurements improves overall assay performance. Nevertheless, depending on the serological cohort, IgG-only often provides sufficient discrimination as IgM contributes mainly in acute sera. In resource-limited settings, it remains feasible to measure only IgG. Our work offers robust benchmarks for the expected performance loss in this scenario and can guide decisions about whether this trade-off is acceptable.

Importantly, ML models, such as GBC and LDA, can be continuously refined as new data becomes available, ensuring that the model remains aligned with evolving immune profiles and epidemiological dynamics. Furthermore, the high degree of multiplexing offered by our assay provides a foundation for future studies to longitudinally monitor immune responses to 15 distinct antigens after infection, vaccination or breakthrough infections.

Finally, the challenges associated with serological cross-reactivity are not unique to orthopoxviruses. Similar diagnostic difficulties arise in other virus families, notably *Flaviviridae* and *Hantaviridae*, where high sequence homology between structurally related antigens can confound traditional serological assays[62,63]. Our ML-assisted multiplex approach could be adapted to these and other contexts to enable high-resolution serostatus classification, improving the specificity and interpretability of serodiagnosis tools across a broad range of infectious diseases. The key strength of our approach is its ability to effectively dissect the cumulative effect of individually weak and overlapping patterns in the immune response, where single antigens fail to distinguish between groups.

In conclusion, our assay combines multiplex serology and machine learning to achieve differentiation between MPXV infection and MVA vaccination. Its high-throughput design and minimal sample processing make it particularly well-suited for large-scale public health surveillance. This enables monitoring of MPXV infections, supports data-driven vaccine strategy decisions, and strengthens outbreak response to emerging viral threats. Our work thus introduces an integrated framework that combines multiplex serology with machine learning. By resolving immune responses to closely related viruses such as MPXV and MVA at high specificity, our assay enables distinctions that classical serology alone cannot achieve. More broadly, our results demonstrate that leveraging high-dimensional serological data through ML can reveal latent patterns in immune responses, paving the way for a new generation of seroepidemiological studies with significantly improved resolution. We envision this strategy becoming a blueprint for differentiating antibody immune responses, especially those that result in cross reactive antibody responses to closely related viruses, such as the flaviviruses, encouraging a shift from single-marker assays toward multiplexed, computationally driven analysis in both research and surveillance contexts.

## Methods

### Serological cohorts and ethics

Ethical approval for this study was obtained from the Berliner Ärztekammer (BÄK) under the references Eth-26/22, Eth-40/22 and Eth-44/22. Samples from individuals in all serological cohorts were collected with informed consent from all participants (pre-immune or MVA group in the acute and validation cohorts) and with approval from the relevant ethics committee (MPXV group in the acute cohort: BÄK Eth-26/22; epidemiological cohort: BÄK Eth-40/22; samples for method comparison: BÄK Eth-44/22). Commercially obtained pre-immune sera were used as research reagents and therefore did not require additional ethical clearance.

Data were pseudonymised and handled in accordance with relevant guidelines and regulations. Where available, metadata included year of birth, MPXV infection status and time of infection, MVA/smallpox vaccination history, and reference assay results (IFA, ELISA, NT)[17].

To establish and benchmark the multiplex assay, we used a comparison cohort comprising sera from PCR-confirmed CPXV or MPXV infections and from MVA vaccines. Sera were selected to cover a broad range of IgG and IgM reactivities. For IFA comparison, 25 sera were tested (20 CPXV, 4 MVA, and 1 Parvovirus B19). ELISA benchmarking included 14 sera from 5 individuals at pre-, post-prime, and post-boost stages of MVA vaccination. We also included 36 sera from 18 PCR-confirmed MPXV cases (≤33 days post-infection), and 21 sera previously tested by IFA for ELISA cross-validation. NT comparison involved 25 sera from 10 MPXV cases tested for VACV-neutralization as described below and elsewhere[17].

For training and testing the ML models, we used two different cohorts: an acute cohort and an epidemiological cohort (Table 2). The acute cohort had 307 sera from 173 PCR-confirmed MPXV cases sampled during acute or early convalescent phases (May–August 2022). Additionally, 54 sera from 18 MVA vaccinees were included across five immunization timepoints (including 17 pre-immune sera). Two pre-pandemic control cohorts (n = 88 each), collected prior to the mpox outbreak in Germany, served as negative controls; one cohort included age data (Table 2).

The epidemiological cohort was collected between April and June 2023 in a high-risk MSM population attending STI clinics in Berlin. Sex/gender was self-reported. Metadata included prior MPXV diagnosis, MVA vaccination (dose count), birth year, smallpox vaccination history, and mpox-related symptoms. Among 859 participants, 59 reported prior MPXV infection, 476 MVA vaccination, and 324 neither infection nor vaccination. Individuals reporting both infection and vaccination (n = 8) were classified as MPXV-infected for model training. The study cohorts reflect the demographic distribution of the 2022 mpox outbreak, which predominantly affected men who have sex with men (MSM).

Sera from the acute cohort were used to develop the assay and were measured unblinded. The epi cohort was processed subsequently in a blinded manner using three bead lots, then unblinded for ML model training. This minimized systematic bias for the epi cohort, though not for the acute cohort.

An independent validation cohort included 50 pre-mpox outbreak sera from a commercial supplier (Central-BioHub, Germany), considered negative for MPXV and MVA. Additional in-house samples included 3 pre- and 35 post-MVA vaccination sera, and 55 post-MPXV infection sera collected through mpox surveillance in Berlin.

## Sex and gender reporting

Information on sex and/or gender was not available for the majority of the serum samples analyzed in this study, as these were either anonymized or provided without associated metadata. Consequently, sex and/or gender was not considered in the study design, and no sex- or gender-based analyses were performed. The lack of this information precluded disaggregated reporting by sex or gender.

## Antibodies and reagents

Suppliers and sources of all antibodies (primary and secondary) used in this work are shown in Supplementary Table 15. The following reagents were obtained through the NIH Biodefense and Emerging Infections Research Resources Repository, NIAID, NIH: polyclonal anti-Vaccinia virus (immune globulin G, Human), NR-2632 (vaccinia immune globulin VIG), and monoclonal anti vaccinia virus (Western Reserve - WR) B5-VACV protein, residues 20 to 275 (ectodomain), (similar to VMC-14 and produced in vitro), NR-429. Samples of antibodies produced in house can be obtained from the authors upon request and availability

## Antigens and quality control

An overview of all antigens used in this work is shown in Supplementary Table 16. Sequences of primers and details of plasmid construction for recombinant proteins generated in this study are provided in Supplementary Table 17. These materials have not been previously published. Plasmids are available from the corresponding author upon reasonable request. Antigens were obtained either through the National Institutes of Health (NIH) Biodefense and Emerging Infections Research Resources Repository (BEI Resources), National Institutes of Allergy and Infectious Diseases (NIAID at NIH), from ProteoGenix (Schiltigheim, France), or custom expressed in E. coli (GenExpress, Berlin, Germany). The following reagents were obtained through the National Institutes of Health (NIH) Biodefense and Emerging Infections Research Resources Repository, National Institutes of Allergy and

Infectious Diseases (NIAID at NIH): vaccinia virus (Western Reserve strain- WR) A27-VACV protein with C-terminal histidine tag, recombinant from baculovirus, NR-22133, vaccinia virus (WR) L1 protein with C-terminal histidine tag, recombinant from baculovirus, NR-21986, vaccinia virus, Western Reserve, A33-VACV protein with C-terminal histidine Tag, recombinant from baculovirus, NR-2623, and vaccinia virus (WR) B5-VACV protein with N-terminal histidine tag, recombinant from baculovirus, NR-22132. Lysates of VACV-infected HEp-2 cells and non-infected cells were produced as described[16]. Briefly, HEp-2 cells were infected with VACV at a multiplicity of infection of 0.2 and incubated for 3–4 days until cytopathic effects were evident and cells detached. Infected and non-infected control cells were lysed in RIPA buffer supplemented with HALT™ Protease Inhibitor Cocktail (Thermo Fisher Scientific, Dreieich, Germany). Protein concentrations were determined using a bicinchoninic acid (BCA) assay (Pierce, Dallas, TX, USA) following the manufacturer's instructions. Lysates were used to coat ELISA plates at 4 µg/mL in 0.1 M carbonate buffer (pH 9.6) overnight at 4 °C. Wells coated with lysate from non-infected cells served as controls for non-specific antibody binding. Samples of recombinant antigens produced in house can be obtained from the authors upon request and availability. Alternative sources for unique recombinant antigens (e.g. ATI-N-CPXV and MPXV-A27) are available from commercial suppliers. The purity and identity of all recombinant orthopoxvirus-specific antigens was checked by Liquid Chromatography-Tandem Mass Spectrometry (see Supporting Methods).

## Bead coupling and multiplex assay workflow

Reagents and preparations of buffers used for coupling MagPlex® Microspheres and performing the multiplex assay have been previously described[64]. In summary, MagPlex® Microspheres (bead regions 7, 8, 15, 20, 26, 33, 36, 38, 42, 48, 52, 54, 55, 59, 67, 76, 81, 82, and 89) were obtained from Luminex® Corporation (DiaSorin, Saluggia, Italy). 1-ethyl-3-(3-dimethylaminopropyl)carbodiimide (EDC) and sulfo-N-hydroxysuccinimide (NHS) were obtained from Thermo Fisher Scientific (Dreieich, Germany). Phosphate buffered saline (PBS) was produced in-house, and other buffer solutions ($NaH_2PO_4$, MES, PBS/TBN, PBS/B, PBS/T) were prepared according to the xMAP® Cookbook. MagPlex® Microspheres were coupled with proteins as outlined in Supplementary Table 15. Briefly, Protein coupling to MagPlex® microspheres was performed using standard EDC/NHS chemistry according to the xMAP® Cookbook (4th Edition, Luminex® Corporation). Briefly, $1.5 \times 10^6$ beads per region were activated with EDC and sulfo-NHS in $NaH_2PO_4$ buffer (pH 6.2) for 20 min and subsequently washed in MES buffer (pH 5.0). Proteins were coupled for 2 h at room temperature. After blocking in PBS/TBN for 30 min, beads were washed and resuspended in PBS/TBN, and concentrations were adjusted to 1,000 microspheres/µL. Successful coupling and retained antigenicity were shown by specific binding of antibodies targeting one of the recombinant proteins[65] as well as broad recognition of all specific antigens by vaccinia immune globulin (VIG, BEI Resources, NR-2632) as described in the Supporting Methods (Supplementary Fig. 6).

All sera, except the commercial sera used for the assay validation, were inactivated as described[17]. Briefly, we used a combination of detergent and heat treatment that included a 30-minute incubation at 56 °C with 0.5% Triton/0.5% Tween. To determine IgG and IgM antibodies against the orthopoxviral proteins in the bead-based multiplex assay, each serum sample was diluted (final) 1:100 and 1:1000 in PBS/B (PBS + 1% BSA) assay buffer, before addition of the multiplex bead mix containing all antigen-coupled MagPlex® beads to be tested. On each plate, five dilutions of VIG in a 1:4 dilution series starting at a 1:500 dilution was included as well as a blank control containing buffer only, and one negative serum measured in a 1:100 and 1:1000 dilution. A layout using only the first six columns of a 96 well plate of all sera and

standards was mirrored in the second set of 6 columns so that both IgG and IgM detection could be performed on the same plate. The multiplex assay was performed as previously described (Supporting Information)[64]. Briefly, multiplex bead mixes containing 1,000 beads per region for each antigen were freshly prepared for each assay. Fifty microlitres of bead mix and 50 μL of diluted serum or standard were added per well of 96-well microplates and incubated for 1 h at room temperature on a plate shaker (750 rpm). Plates were washed three times with PBS containing 0.1% Tween 20 using a magnetic plate washer with 1-min soak times. PE-labeled secondary antibodies (goat anti-human IgG or donkey anti-human IgM, 1 μg/mL in PBS with 1% BSA) were added and incubated under the same conditions, followed by identical washing. Beads were resuspended in 100 μL PBS and fluorescence was measured on a Bio-Plex 200 system (Bio-Rad) at normal detector gain, collecting ≥ 50 beads per region. Raw data were exported in XLSX format for downstream processing in R.

### Reference assays

The three reference methods for validation of the multiplex assay, the virus NT, the IFA, as well as an in-house indirect ELISA were performed as published[17]. Briefly, Neutralizing antibody titres were determined by a microneutralization assay using Vero E6 cells. Serum samples were heat-inactivated and serially diluted (1:10–1:320) in DMEM containing 10% FCS, mixed with an equal volume of virus stock, and incubated for 1 h at room temperature. The mixtures were then added to Vero E6 cell monolayers in 96-well plates and incubated for seven days at 37 °C and 5% $CO_2$. Cytopathic effects (CPE) were assessed microscopically, and neutralizing titres were defined as the highest serum dilution preventing CPE in ≥50% of wells.

IFAs were performed using OPXV-infected HEp-2 cells (VACV Lister-Elstree or CPXV HumGri07/1) as antigens. After acetone fixation, slides were incubated with heat-inactivated sera (IgM sera pre-treated with Mastsorb Absorbens) followed by FITC-labelled anti-human IgG or IgM and counterstained with Evans Blue. Fluorescence was evaluated by microscopy.

In-house ELISAs were conducted with UV-inactivated VACV-infected HEp-2 cell lysates as antigens and lysates of uninfected cells as negative controls. Sera were tested in serial dilutions (1:100–1:6400) for IgG or IgM using HRP-conjugated secondary antibodies and TMB substrate. Absorbance was measured at 450 nm (reference 620 nm) using a microplate reader.

### Data processing and normalization

All data analysis was done using the statistical programming language R (version 4.3.0)[66]. Scripts and data used to generate all figures and supporting information are shared via GitHub (https://github.com/RKI-ZBS/bead-based_multiplex). The following R packages were used: broom (v1.0.7), caret (v7.0-1), circlize (v0.4.16), ComplexHeatmap (v2.18.0), corrplot (v0.95), cvms (v1.6.3), data.table (v1.16.4), drc (v3.0-1), GGally (v2.2.1), ggbeeswarm (v0.7.2), ggimage (v0.3.3), ggpubr (v0.6.0), ggthemes (v5.1.0), gplots (v3.2.0), mcr (v1.3.3.1), plyr (v1.8.9), pROC (v1.18.5), rio (v1.2.3), rstatix (v0.7.2), rsvg (v2.6.1), scales (v1.3.0), stringr (v1.5.1), tidyverse (v2.0.0), and yardstick (v1.3.2). In all box-and-whisker plots shown, the lower and upper hinges represent the first and third quartiles (25th and 75th percentiles). Whiskers extend to the smallest and largest values within 1.5 × IQR (interquartile range) from the hinges. Data beyond the whiskers are considered outliers and plotted individually.

To normalize binding responses across plates, a standard curve of five VIG dilutions was included on each plate. VIG IgG concentrations ranged from 10,000 μg/mL to 39 μg/mL in five 1:4 serial dilutions. For each orthopoxvirus-specific antigen, a 4-parameter logistic regression model was fitted to $log_{10}$-transformed mean fluorescence intensity (MFI) versus $log_{10}$ IgG concentration using the drLumi package[67], with buffer-only controls used to constrain the curve's lower bound. Serum

dilutions (1:100 and 1:1000) were interpolated against the standard curve per antigen. Limits of quantification (LOQ) were defined by a coefficient of variation <30% within the standard curve. Measurements outside the LOQ were excluded. Final antigen-specific concentrations were determined as the mean of both dilutions if within LOQ, the 1:1000 dilution if the 1:100 was above LOQ, the 1:100 dilution if the 1:1000 was below LOQ. If both measurements were outside the LOQ range, antigen-specific minimum or maximum values were assigned accordingly.

Due to the absence of an IgM standard, IgM data were also quantified using the IgG standard curve to minimize inter-plate variability leading to quantified results (in ng/mL, reported as Multiplex quant.). For antigens, against which VIG is weakly reactive, quantified values are hereby higher, as higher concentrations of VIG are needed to reach certain MFI levels in comparison to antigens, against which VIG is highly reactive. Hence, the overall dynamic range is indicative of antigenicity, whereas the absolute values represent the reactivity of VIG against the antigens.

### Assessment of reproducibility measurement

Reproducibility was assessed by independently quantifying antibody levels in 168 serum samples (39 from the epidemiological cohort, 129 from the validation cohort). For each antigen, the coefficient of determination ($R^2$) was calculated by linear regression between the two independent measurements. Additionally, the F1 score was determined for each antigen to evaluate the agreement in binary classification as seropositive or seronegative. High reproducibility was observed for highly reactive antigens, as indicated by $R^2$ values and F1 scores approaching 1 (see Fig. S13). Agreement was generally higher for IgG than for IgM measurements, while antigens with lower immunoreactivity exhibited reduced concordance.

### Machine learning implementation and evaluation

The goal of the classification task was to assign each serum sample to one of three serogroup categories, based on multiplex serology data: Pre–no known MPXV infection or MVA vaccination, MVA–history of MVA vaccination, MPXV–PCR-confirmed (acute cohort) or self-reported (epi cohort) MPXV infection. Stratification by childhood smallpox vaccination or MVA dose number was not performed, as this data was incomplete.

Machine learning models were implemented in Python using the scikit-learn library[68], with fuzzy rule-based classification (FRBC) implemented via the skmoefs package[69]. Input data consisted of normalized (min–max scaled) IgG and IgM measurements from the acute and epidemiological (epi) serological cohorts. All orthopoxvirus-specific antigens were included as features, except for L1-VACV and M1-MPXV, which were excluded due to weak antibody responses and inconsistent bead-coupling reproducibility–factors that inflated model performance during preliminary testing.

Six models were trained using three algorithms, Random Forest (RF), Gradient Boosting Classifier (GBC), and Linear Discriminant Analysis (LDA), on IgG-only or combined IgG/IgM data from the acute cohort, the epi cohort, or both combined (see Fig. 1). We selected RF and GBC to represent two common, complementary ensemble learning approaches: bagging and boosting, respectively. These models are consistently reported to perform well across biomedical datasets and offer interpretable feature importance rankings[70–72]. RF served as a strong baseline model due to its stability and interpretability, while GBC was chosen for its ability to capture complex, non-linear interactions, a key requirement in distinguishing overlapping serological profiles. Given the non-normal, often skewed feature distributions (Supplementary Fig. 12, Shapiro–Wilk tests), we prioritized tree-based models (RF, GBC) that capture non-linear interactions; LDA served as a linear baseline to benchmark performance under stronger distributional assumptions (multivariate normality

and equal covariance across classes), assumptions that are often violated in immune response datasets due to skewed, heterogeneous, or bimodal antigen distributions. XGBoost was not included to avoid redundancy with GBC. Lasso regression was considered less appropriate due to its limited utility in multi-class settings and assumption of linear separability. All machine learning models were trained using the same set of antigen features and serum samples, with no variation in input data across models. To evaluate model performance and robustness, we applied stratified 5-fold cross-validation, repeated three times (15 runs per model). This approach ensures that each model is exposed to the same training and testing conditions, allowing direct comparison of classification performance based solely on differences in model architecture. Cross-cohort predictions were also conducted to assess model transferability. To this aim, models were trained on the entire source cohort and tested on the entire target cohort. Evaluation metrics included macro-averaged F1 score, accuracy, precision, and recall, as previously described[73].

To determine the contribution of each antigen to the overall model performance for GBC, we performed recursive feature elimination using the combined IgG and IgM dataset. To this aim, we first calculated the permutation importance of each antigen on the complete dataset to assess its impact on the model's predictive power. The antigen with the smallest permutation importance was then sequentially removed. Mean F1 scores (macro-averaged) from 15 cross-validation runs were calculated for models evaluated on either the combined acute and epidemiological or the independent validation serum cohorts, using all antigens (baseline) or after sequential removal of the antigen with the smallest permutation importance. By cumulatively excluding the least informative antigens, we benchmarked model robustness and identified a minimum antigen set that maintained classification performance comparable to the baseline model.

### Ensemble learning algorithm

For the independent validation cohort, we generated ensemble predictions by aggregating 30 model instances per algorithm (15 trained on IgG-only and 15 on combined IgG/IgM; stratified 5-fold CV repeated three times on the combined training cohorts). Final labels were assigned by majority vote within each algorithm (GBC, LDA, RF). To quantify uncertainty, we bootstrap-resampled the validation set ($n = 2000$ iterations, with replacement) to estimate mean performance metrics (macro-F1, precision, recall, sensitivity, specificity) with 95% confidence intervals. For comparability, the same procedure was applied to the combined establishment dataset (acute + epidemiological), and results are reported as means with 95% confidence intervals. A combined ensemble (ensemble serostatus) was evaluated alongside the single algorithms. In this ensemble, LDA was applied to seronegative samples and GBC to seropositive samples, with serostatus defined by the Delta-VACV threshold. To test whether performance improves when uncertain calls are removed, we applied a confidence threshold of 0.5 (ensemble probability; LDA posterior), retaining predictions with probability > 0.5 and flagging lower-probability calls as low-confidence; results are shown with and without this filter.

### Performance of single antigen classifiers in comparison to ML algorithms

To assess the performance of single antigen classifiers versus trained machine learning (ML) models, we conducted bootstrap analyses ($n = 2,000$ iterations) on ML-based ensemble predictions (GBC) or antigen-based predictions on both the combined cohort ($n = 1,260$) and the independent validation dataset ($n = 143$). For the single antigen approach, sera were classified as either MPXV-infected based on ATI-N-CPXV reactivity or the ratio of binding to A35-MPXV/A33-VACV above

the threshold defined by ROC analysis, or as seropositive based on E8-MPXV or B6-MPXV reactivity. To enable direct comparison, predictions from the GBC algorithm were recoded to match the binary classification used. Furthermore, predictions were combined to generate the three-class output produced by the GBC algorithm based on their reactivity profiles: pre-immune (E8 or B6 negative, regardless of ATI-N reactivity or A35-MPXV/A33-VACV ratio), MVA-vaccinated (E8 or B6 positive, ATI-N-CPXV or A35-MPXV/A33-VACV ratio negative), or MPXV-infected (double positive).

### Robustness of ML-based classification of multiplexed serological data

To assess the robustness of ML-based classification, we systematically evaluated the GBC algorithm trained on the combined dataset. The model was trained and tested either with the complete antigen panel or with one or more of the most influential antigens, ATI-N-CPXV, E8-MPXV, and D8-VACV, excluded. Specifically, we compared the baseline model (all antigens included) to models with individual antigens removed, both ATI-N-CPXV and E8-MPXV removed, and with all three key antigens excluded. Mean F1 scores and 95% confidence intervals of ensemble predictions were calculated as described above.

### Technical Use Disclosure

ChatGPT-4.0 and 5.0 were used solely for language editing, including grammar correction, typesetting, and improving readability. No scientific content was generated by the model.

### Reporting summary

Further information on research design is available in the Nature Portfolio Reporting Summary linked to this article.

## Data availability

The multiplex serology and machine-learning data generated in this study have been deposited in the Figshare repository under accession code https://doi.org/10.6084/m9.figshare.26391226. The raw individual-level data containing metadata are protected and are not publicly available due to data-privacy regulations. Processed data underlying all figures and tables, as well as the datasets used to train and validate the machine-learning models, are provided in the Source Data file. The corresponding analysis scripts are available at https://github.com/RKI-ZBS/bead-based_multiplex. Source data are provided with this paper.

## Code availability

Input data, metadata and R code to reproduce all figures and tables is contained within the following GitHub repository: https://github.com/RKI-ZBS/bead-based_multiplex. Python code for the implementation of the ML code is available here: https://github.com/ZKI-PH-ImageAnalysis/Mpox_Multiplex_Assay.

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

## Acknowledgements

This work was supported by grants from the German Federal Ministry of Health. The authors thank Josephina Lambrecht and Silvia Muschter for excellent technical assistance and Marcia Triunfol for excellent assistance in scientific writing.

## Author contributions

R.S., F.T., F.B., and A.S. contributed equally. R.S., F.T., F.B., A.S., M.L.N., T.R., M.G., M.S., J.M., and D.S. designed and/or performed the experiments. S.A., D.B., and N.K. performed and/or supervised the ML analysis. S.A., D.B., M.S., H.W.M., and D.S. analyzed the data. K.J., U.K., U.M., N.F., A.M., N.K., A.N., and D.S. conceived the study or parts of the study, B.G.D. and K.L obtained funding and supervised the project. D.S. and R.S. wrote the paper with input from all other authors. All authors read and approved the final manuscript. This work was supported by a pilot project and special research funding ("9-Punkte-Plan") provided by the German Federal Ministry of Health (Bundesministerium für Gesundheit, BMG) to the Robert Koch Institute (RKI). Funding was awarded to F.T., D.S., and B.G.D. as part of institutional research support.

## Funding

## Competing interests

The authors declare no competing interests.

## Additional information

[1]Highly Pathogenic Viruses (ZBS 1), Centre for Biological Threats and Special Pathogens, German Consultant Laboratory for Poxviruses, WHO Collaborating Centre for Emerging Infections and Biological Threats, Robert Koch Institute, Berlin, Germany. [2]Biological Toxins (ZBS 3), Centre for Biological Threats and Special Pathogens, Robert Koch Institute, Berlin, Germany. [3]Image Analysis (ZKI-PH 3), Centre for Artificial Intelligence in Public Health Research, Robert Koch Institute, Berlin, Germany. [4]HIV/AIDS, STI and Blood-borne Infections (FG 34), Robert Koch Institute, Berlin, Germany. [5]Measles, Mumps, Rubella, and Viruses Affecting Immunocompromised Patients (FG 12), Robert Koch Institute, Berlin, Germany. [6]Centre for Artificial Intelligence in Public Health Research, Robert Koch Institute, Berlin, Germany. [7]Medical University Lausitz - Carl Thiem (MUL-CT), Cottbus, Germany. [8]These authors contributed equally: Rebecca Surtees, Fridolin Treindl, Shakhnaz Akhmedova, Denis Beslic. ✉e-mail: NitscheA@rki.de; SternD@rki.de

