## [Peer Review file · Nature Communications]

Machine learning-supported framework for the classification of mpox infection and MVA immunization from multiplexed serology data

Corresponding Author: Dr Daniel Stern

Version 0:

Reviewer comments:

Reviewer #1

(Remarks to the Author)
To whom it may concern,

RE: Review of Nature Communications manuscript NCOMMS-24-46584 entitled "Differentiation between mpox infection and MVA immunization by a novel machine learning-supported serological multiplex assay"

The above paper described the training and use of different machine-learning algorithms to analyse antibody data for a number of immunodominant Orthopoxvirus antigens in several relevant cohorts. The work focuses on building and testing a model to distinguish between these different cohorts (principally naïve, post-vaccination and post-infection), which is of timely relevance due to the concurrent clades I and II outbreaks. With smallpox vaccines becoming more widely used in response to the outbreaks, the cross-reactivity of antibody responses to Orthopoxviruses is a growing challenge that warrants investigation. Data analysis models capable of teasing apart responses to different Orthopoxviruses will be key in diagnostics, serosurveillance and improving the next generation of Orthopoxvirus vaccines. The manuscript is written and presented well throughout, and I believe analysing serological data for MPXV in this way and the insights provided to be original and of interest to the field.

Given the complexity of antibody responses to MPXV infection/vaccination and the increasing use of multiplexed assays more generally, tools able to digest and interpret the large amounts of data produced, such as those described in this paper, are of importance and noteworthy. The differences described between the antibody responses of the two panels (acute and epi) and the skew of post-vaccination responses to VACV antigens were of particular interest. Potential explanations of the mechanisms behind these differences would enrich the discussion. The paper also highlights the benefits of using multiplexed assays for assessing antibody responses to multiple antigens simultaneously.

The conclusions made in this paper are well supported by the results presented therein and no additional evidence is required. Due to a lack of personal expertise, I am unable to comment on the machine learning aspect of the methodology in much specificity. However, this side of the paper was very well explained and in sufficient detail for a layperson to understand the results of this process. All other aspects of the methodology are sound, of the expected standard and in sufficient detail for the work to be reproduced.

Please find minor comments below, thank you.

MINOR COMMENTS:

• THROUGHOUT:

- o Proteins are referred to using the gene names (e.g. B6R) and not the protein name (e.g. B6).
- o It is unclear which exact proteins are being referred to in the text and figures. For example, in Figure S10 graphs are labelled with A33 and A35. It is assumed from the methods that A33 is the vaccinia protein and A35 is the MPXV protein. However, this is worth clarifying throughout to avoid confusion as VACV, CPV and MPXV express different but similar named proteins A33 and A35.

- INTRODUCTION LINE 125:

- o The paper referenced describes post-MPXV sera having preferential binding to the MPXV proteins A29, A35 and B6 over their VACV homologues. In contrast, post-MVA-BN sera showed preferential binding to VACV A27, A33 and B5 over their MPXV homologues. Importantly, both vaccination and infection produced antibodies to these proteins (less so for vaccination and MPXV A29 / VACV A27). However, it was suggested that these differences in binding specificities (i.e. the ratio of responses to homologues) may be used to differentiate between infection and vaccination.
- o Comparisons of VACV L1 and MPXV M1 were not done as suggested in the manuscript.
- o Separately to this, the paper referenced showed that antibody responses to proteins MPXV A27 (N-terminal of A-type inclusion body protein) and MPXV M1 were significantly different in infected individuals versus those who had been vaccinated. Infected individuals produced a measurable antibody response to MPXV A27 whereas vaccination did not.
- o In contrast, vaccinated individuals mount robust responses to MPXV M1 whereas infected individuals have weak/variable responses.
- o It was this difference in response to MPXV A27 that was used to differentiate between infection and vaccination.
- o For accuracy, please reword to 'it has been demonstrated that polyclonal antibodies produced post-infection or immunization exhibit differential binding to some of these antigens, namely A33 and B5. An observation suggested for serological differentiation by ELISA.'

- RESULTS LINES 237-240:

- o Creep of discussion into the results.

- DISCUSSION LINE 384:

- o Similar results were also demonstrated in <https://www.nature.com/articles/s41467-023-41587-x>

- DISCUSSION

- o The limitations of this work have not been discussed sufficiently.

(Remarks on code availability)

I do not have the appropriate expertise to review the code provided.

Reviewer #2

(Remarks to the Author)

(Remarks on code availability)

Reviewer #3

(Remarks to the Author)

Review of "Differentiation between mpox infection and MVA immunization by a novel machine learning-supported serological multiplex assay" for submission to Nature Communications by Yates et al.

This study presents a promising method for distinguishing immune responses from mpox infection and vaccination through machine learning (ML) analysis of a multiplexed serological assay. A significant strength of the study is the use of 15 poxvirus protein panels in conjunction with ML to interpret complex data from the multiplexed assay. This approach addresses the critical need for more accurate serological monitoring, particularly for zoonotic diseases like mpox. The authors effectively demonstrate that integrating ML with multiplexed assays enhances the accuracy of tracking complex antibody responses, offering a valuable tool for future epidemiological studies and monitoring.

In summary, this study is important in validating the effectiveness of multiplex assays and computational analysis for serological epidemiological monitoring of infectious diseases. The solid approach, well-structured methodology, clear metrics, and the study's epidemiological relevance are commendable. Additionally, the sample set is well-organized, and the author's efforts in this research are highly respected.

The comments below are to be published.

-The novelty of the assay and ML analysis needs further clarification. Although the design of the multiplexed protein panel is impressive, and the approaches are well-conceived, similar methods have been applied to COVID-19 and other diseases. The authors should explicitly highlight their study's unique aspects and contributions compared to existing approaches.

-The manuscript demonstrates the effectiveness of the bead-based multi-antigen panel in distinguishing between mpox-infected and vaccinated individuals. It also shows the suitability of ML for handling the data. However, the manuscript lacks a detailed discussion on how ML specifically improves performance over traditional methods. The authors should include a more thorough explanation of how ML enhances assay performance and address any limitations that might be present in multiplexed assay results when analyzed without ML.

-Given the complexity and non-linear nature of the results from the multiplexed panels, a deeper exploration of the data analysis techniques is needed. In such cases, neural networks might offer advantages over traditional ML algorithms in capturing complex, non-linear patterns. The authors should consider discussing why they chose ML algorithms over neural networks and whether they explored deep learning alternatives.

-The study may not present the blinded sample set. While five-fold cross-validation is valid, it may not always reflect real-world, unseen data performance. The authors should include results from blinded sample testing to ensure robustness and confirm that the model generalizes well beyond the cross-validation.

In the abstract and introduction, it would be helpful to briefly mention the number of samples tested with simplified information to provide the study's scale.

Minor comments

-The link for reference 12 is incorrect.

-In line 79, "two-fold" is unclear and should be used as another term.

-Figures 1b and S10 appear to be redundant. The IgM results in Figure 1b and the IgG results in Figure S10 should be removed to avoid duplication.

-Please check the labels in Figure S9, as there are some typographical errors. I also noticed similar errors in other figures, so a careful review is needed.

-In Figure 1a, consider using different colors for the lines. The current blue and green lines are hard to distinguish and too similar to the IgM/IgG color.

-Please switch Figures S14 and S15, as Figure S14 is mentioned after Figure S15 in the manuscript.

-Figures S18-S20 and S22-S23 should be replaced with higher-resolution versions. The labels in the plots are not clear.

-The term "PBS/T buffer" in the Methods section (line 582) is unclear. It may refer to the incubation buffer, but this should be clarified.

(Remarks on code availability)

Reviewer #4

(Remarks to the Author)

In "Differentiation between mpox infection and MVA immunization by a novel machine learning-supported serological multiplex assay" Surtees et al. applied multiplexed assays to disentangle the humoral immune response against mpox infections and vaccinations. The work is timely and interesting, but several major concerns have decreased our excitement about this work.

1) The manuscript is very long and descriptive in nature. Too many unnecessary details consume the attention of what is actually important. Overall, this has led to a very bloated text. A revision should, therefore, include a substantial rewriting and narrowing of the key outcomes. This will decrease the risk of "data dumping".

2) Analyzing the data in all directions is an interesting exercise, but it does not increase the value of the outcome. A clear strategy and clean study design need to be defined and explained before building a useful classifier. Avoid any side stories that could distract from the main focus. To us readers, the work appeared as if they had simply included all results from their data analysis. This is not impressive but distracting. We expect from authors of Nature Communication submissions the ability to moderate and guide readers, rather than leaving them with the data. We also noted that single results were sometimes linked to several figures and tables (both main and supplementary). Jumping between different sub-studies makes it difficult to find what the significance of this work actually is.

3) The multi-parameter assay concept as such is neither novel nor has their model and approach been proven to be informative in independent data sets. The recent pandemic has inspired the community to develop several similar approaches and concepts. A new literature survey will reveal such efforts.

4) Differentiation of infection and vaccination depends on several factors, including the differences between vaccine and viruses, time since infection and vaccination as well as unrelated events interfering with the readout. This must be investigated.

5) It remained unclear to us if different antigen panels were used for the different study sets.

- 6) Please explain better why retrospective differentiation of mpox vaccination from infections is of value. Add a demographic table of the two study sets into the main text.
- 7) Line 104ff discusses the classification performance, but the authors do not explain why. Is it the sensitivity of the assay, waning of Ab levels, immunocompromised donors, or distribution of the data?
- 8) Line 151ff: explain “complex Delta antigen”.
- 9) Line 152 on Ab levels: It is important to remember that the reported MFI values are a combination of factors and not only Ab levels. The authors mention polyclonality of the immune response but should add that this also creates Abs of different affinities. MFI is not only abundance.
- 10) Line 155ff: Using other related correlation methods does not provide independent evidence. It’s a circular argument.
- 11) Line 160ff: The authors have no evidence about the immunogenicity, as their observations could be driven by technical biases that favor one antigen over the other. Neutralization assays or other means of normalizing between antigens must be added to support such claims.
- 12) Line 179: Explain how participants “self-reported mpox infection”.
- 13) Line 185ff: The manuscript in its current form reports on applying different ML methods, but it does not justify their selection over other models and does not benchmark them against less complex statistics.
- 14) Line 196: Explain “presumed childhood infection”.
- 15) Line 205: Explain “simple main effects”.
- 16) Line 207ff: It is confusing to read about different selections of antigens (= panels) and sample sets. Did they use different sets of beads? Did they analyze the samples together? Was the analysis done blinded? This needs to be validated in a new sample set from another collection.
- 17) Line 211ff: Explain why the IgM levels were lower.
- 18) Line 219ff: Explain the sentence “However, differentiation between... “. It does not make much sense as is.
- 19) Line 259ff: The statement about batch effects raises concerns about the overall quality of the data (precision, reproducibility, reliability,...). Add a PCA or UMAP overview, plus a metrics table (Excel sheet) to summarize the assay performance and batch variability. Explain how the randomization across analysis batches was handled to avoid any upfront bias that cannot be accounted for downstream.
- 20) The performance of data models is highly dependent on what the classifier was trained on. The less prototypical and accurate that data is, the less selective and precise its application will be. Interchanging and reusing data for such purposes is not a sustainable practice. It must be much clearer how the authors used their data for such purposes and any reuse of data is discouraged. The authors need to decide on one approach and not flip their model around between the different data sets to find a story worth telling. See also line 291 and explain what they mean by “homologue panel”.
- 21) Line 273: Please explain why they “did not further stratify...”.
- 22) Line 274: Explain which data in “...did not have sufficient data on...” they refer to.
- 23) Line 280: Explain why “5-fold” and not 100-fold.’
- 24) Line 329ff: Explain the sentence “Here we found...classifications...were increased...”.
- 25) Line 338ff: The statement “...excluding low confidence predictions, which also have a higher rate of misclassifications...” is another example of circular reasoning.
- 26) Line 340 is one of the first to introduce the term “cut-off value,” but there is no explanation of what that cut-off is and how others could make use of this for clinical decision-making. It raises the question of how the MFI data was treated and if it was assessed for the criteria needed to fulfill the modeling requirements. Would similar outcomes be reported if the MFI were to be converted into binary classifiers?

In essence, the paper is based on machine learning, so one can expect some complexity in the data analysis approach and data representation. However, it is somehow difficult to understand which is the final panel of antigens and why it performs best in classifying certain phenotypes. Despite its timing, it is confusing for us that IgG, IgM, and IgG+IgM data are discussed interchangeably in different contexts. Whatever model is the most suitable, there needs to be an explanation for the observations. There is too little discussion about the clinical implications and interference for other events between vaccination/infection and sampling.

Other comments:

The authors should check abbreviations/acronyms throughout the manuscript because some are not cited. e.g. IMV and EEV in Tab 1

Terms like good, better, higher, and largely ... are imprecise and must be defined.

The figures include many panels, and the axis, and p-values are difficult to read (e.g., Fig 3). The authors may consider reducing the panels and selecting the most interesting plots.

Fig 1. Which samples were used to correlate the multiplex bead assay with the IFA and NT assays? Fig 1a. How were the intervals in x-axis selected for plots 3-6?

Fig 1b and line 152: the authors refer to "highest autoantibody levels" detected for some of the tested antigens, but it seems that all antigens reach similar levels in Fig 1b.

Fig 2 represents that the acute and epidemiological sample panels were tested for either IgG or IgG+IgM, but it seems to me that the results report either IgG OR IgM (see Fig 3)

(Remarks on code availability)
No code was reviewed at this time.

Reviewer #5

(Remarks to the Author)
I co-reviewed this manuscript with one of the reviewers who provided the listed reports. This is part of the Nature Communications initiative to facilitate training in peer review and to provide appropriate recognition for Early Career Researchers who co-review manuscripts.

(Remarks on code availability)

Version 1:

Reviewer comments:

Reviewer #1

(Remarks to the Author)
All previous concerns have been addressed in the revised manuscript. Thank you.

(Remarks on code availability)

Reviewer #2

(Remarks to the Author)
I co-reviewed this manuscript with one of the reviewers who provided the listed reports. This is part of the Nature Communications initiative to facilitate training in peer review and to provide appropriate recognition for Early Career Researchers who co-review manuscripts.

(Remarks on code availability)

Reviewer #3

(Remarks to the Author)
Thank you for the thorough and thoughtful revisions. I am satisfied with the changes made in response to the comments. The manuscript is significantly improved in clarity, robustness, and presentation. I support the acceptance of the revised version.

(Remarks on code availability)

Reviewer #4

(Remarks to the Author)

We thank the authors for their substantial efforts to improve the manuscript in terms of length, structure, and figures. The overall story has become somewhat clearer, and the flow of the results is easier to follow.

However, and despite this major revision, our level of excitement about the insights derived from the work has not increased significantly. We do not have any stakes in this area of research, but as readers and experts in proteomic studies of the immune system, the intermixed messaging about ML/AI and multiplexed serology left us with the question about the impact of the work and how it would change the community's approach to adapt to their observations.

Following the title "Differentiation between mpox infection and MVA immunization by a novel machine learning-supported serological multiplex assay", our understanding was that the work consists of the task to (1) differentiate mpox infection and MVA immunization, (2) use multiplexed serology assays, and (3) use ML to extract a meaning model from the data. In the abstract, however, the outcome is described as "our approach enables precise immune profiling and enhances the accuracy and utility of mpox serosurveillance." To us, there is a disconnect between the purpose of immune profiling (= following multiple immune system components over time) and the need for differentiation between infection and vaccination (= cross-sectional analysis). Or do the authors mean that the versatile antigen panel + ML can be used for both by keeping the antigen panel and tuning the data by ML?

Re comment 9) We had previously commented on the absence of evidence about antigen immunogenicity (comment 11) and the authors state that they agree. Still, in the new sentence at line 113ff, in response to comment 9, they state again "To further assess the immunogenicity of the selected recombinant antigens..."

Re comment 13) We asked why the authors included specific models in their investigation and not why they prioritized GBC in the end. We would have wished for a more focussed investigation and description of their assumptions and then benchmarking these against classic models like lasso or even XGBoost. The authors use random forest (RF) as a more widely used and pre-ML models, which is good, but there is a need to use additional classical approaches and take RF as a baseline to present the improvement in performance of GBC (or LDA), such as in differences of F1 scores. This should be reported also for the sample set used for validation (Table 3).

Before such an exercise, we suggest that the authors revisit all the fundamental requirements for the chosen models. As they write in lines 334ff about the distribution of the data as a criterion for the models, it is necessary that the model assumptions match with the actual data. Most models are not specifically built for serology data but can we really expect a normal distribution in immune response data (or would forcing a distribution onto the data bias the outcome)? We would rather expect the data to be skewed by tails indicating infections (upwards) or waning (downwards), or even observe bimodal distributions (\pm vaccination). Hence, we have some doubt that these aspects have been taken into account to a degree necessary for the robustness of the ML claims.

As Figs 2 + 4, we understand the idea behind evaluating, training, and testing the models in the acute and epi sample sets interchangeably, but remain uncertain about what the differences are between the models. Do they include a different sub-panel of the antigens or different weights?

Also, we need to know more clearly what the added accuracy of this study is over less elaborated approaches, such as an ELISA assay based on two proteins. Without such reference, it remains unclear whether the approach taken is necessary. That said, the ML-based analysis seems to demonstrate that some improved classification performance can be achieved compared to using the single antigen ATI-N, but what about using two variables: E8 and ATI-N (see Fig. 2b-c, and main antigen branches of Fig. 5a)?

Again, we remain skeptical that an improvement of this magnitude justifies the use of such a broad antigen panel and complex analytical approach in a real-world setting, even in epidemiological studies. As stated in lines 280ff, "Among all antigens, ATI-N-CPXV performed best in distinguishing MPXV-infected individuals, achieving an F1 score of 0.83. Binary classification using GBC achieved slightly better results, with an F1 score of 0.85 and higher specificity (0.95 vs. 0.91 for ATI-N-CPXV alone; Table S13)." We really appreciate this transparency, but it is still unclear to us if a F1 score gain of 0.02 is striking enough to justify the efforts.

We also remain uncertain about the necessity of the multiplexed serology over single or dual analyte assays, and how others should use the presented findings. In lines 353ff, the authors suggest that "Future work should consider incorporating additional antigens with higher specificity and explicitly stratifying models by breakthrough infection status and childhood immunization history." To us, this reads as if the current approach is still insufficient, and better antigens and assays should be designed. There is also a need to reference the outcome of this study to other efforts attempting to achieve a similar outcome.

It remains to be seen if dual detection of IgG and IgM is needed for all studies. While the high-throughput platform allows testing of many antigens across many samples with minimal sample consumption, it remains a costly setup. It would be valuable to know whether the authors have assessed the individual contribution of each antigen to the model and whether a reduced panel could achieve similar performance.

Re comment 16) We had asked to use the term "panel" only for the assay and set of analytes used, not to mix it with and confuse it with the different sets of samples used. However, there remains inconsistency in adapting those changes to more clearly differentiate assay and sample aspects (including the figures).

The message of Fig 3 is unclear to us and the use of the term panel instead of study sets adds confusion. What is the difference between the three double-boxes in the parameter section - aren't they using the same assay panel for all sample sets?

Re comment 19) The new paragraph clarifies the quality control work. This paragraph could be included in the method sections or in results, maybe in a shorter and modified version to fit the section's purpose. However, to show the reproducibility of the measurement, we would rather suggest checking the level of agreement of the 2 experiments in classifying positive and negative control samples. In Fig. 1, which could also be relocated to the supplementary to keep a focus, the data clouds are rather narrow for some of the antigens, making it difficult to appreciate the actual magnitude of the variation between the two different experiments on different antigens. Antigens like ATI-N, B6 and E8 for instance, show a number of samples visibly deviating from the trajectory.

Re comment 26) We asked for a cut-off value that can be used by others. It seems the term is used here solely for the model and not as a value that can be adapted elsewhere (e.g. $MFI > 2000$ or $[c] > 1 \text{ ng/ml}$...). In this regard, we found that the authors have invented terms like "Multiplex Delta Quant" (Fig 1) instead of simplifying the message in units that are translatable to other studies (like $\mu\text{g/ml}$ or else). This continues in Fig 2 (Min Max norm.). It remains unclear to us how others could use these features in their investigations.

Re comment 28) We refer to other viral infections or vaccinations (e.g. influenza) that could influence the data due to molecular mimicry or assay cross-reactivity.

We also noted that the authors do not use any detergent in their assay buffer. Can they please justify why they omitted these very common additives? Is there no concern about unspecific binding?

Re comment 30) We have also requested to avoid relative terms, like high or low, but already in the abstract, we came across "... independent cohort (n = 143) maintained high classification ...". If we read correctly, the F1 score decreased to 0.77 (Table 3), and this should be stated.

(Remarks on code availability)

The code was not assessed during this review.

Reviewer #5

(Remarks to the Author)

(Remarks on code availability)

Version 2:

Reviewer comments:

Reviewer #4

(Remarks to the Author)

We thank the authors for their continuous efforts to improve the manuscript and follow our constructive suggestions.

Overall, the manuscript has substantially improved in text, flow, reasoning, and clarity of the figures.

The benefit of the approach has become a lot clearer. Especially now there is a thorough and transparent comparison with classical methods and their performance (lines 301 ff + Tables S12 + S13). As stated by the authors, AI can support multi-analyte serology for multi-class categorization efforts. Hence, the presented framework showed greater resilience and outperformed classical singular/dual antigen concepts in the presented data. We can foresee an interest in the framework as a mathematical exercise and opportunity for benchmarking in real-life scenarios of other infectious disease phenotypes that might remain more challenging to disentangle (e.g. flaviviruses).

As closing remarks that can be handled with the editorial team, we wish that the authors:

1) modify the title to better reflect these key aspects. One suggestion could be: Machine learning-supported framework for the classification of mpox infection and MVA immunization from multiplexed serology data.

2) add units to all figures presenting data irrespective of the model used. Stating the arbitrary unit [AU] or else will assist future readers to better connect with the presented data.

(Remarks on code availability)
Not evaluated or tested.

Reviewer #5

(Remarks to the Author)

(Remarks on code availability)

Point-by-point response to the reviewers' comments

We would like to express our sincere gratitude to all the reviewers for their time, effort, and the valuable feedback they provided on our manuscript. We have carefully addressed all of the reviewers' comments and made the necessary adjustments accordingly.

In response to the request to further streamline the manuscript, we have made significant revisions to the figures, supporting figures, tables, and text. Some of the questions raised by the reviewers refer to sections or figures that are no longer part of the revised manuscript.

A major revision includes the addition of new measurements from a validation panel. In this section, we utilized our two most robust and high-performing models, GBC and LDA, in an ensemble approach for predictions. We believe this addition significantly enhances our manuscript, and we are grateful to the reviewers for suggesting these additional experiments.

We are confident that the revised manuscript is now clearer and more focused on the key issues. Moreover, the inclusion of the new measurements has substantially strengthened our manuscript. It demonstrates not only the transferability of our models to new datasets but also highlights areas where our assay can be further improved. As more measurements will become available in future work, we anticipate that training and testing the ML models on this specific subset will be beneficial.

Responses to Reviewer #1

1. Proteins are referred to using the gene names (e.g. B6R) and not the protein name (e.g. B6).

Authors' answer: We have removed the ORF notation (L and R) from all the proteins mentioned in the manuscript. All proteins' names now follow the example cited by Reviewer #1, for instance B6R is now B6.

2. It is unclear which exact proteins are being referred to in the text and figures. For example, in Figure S10 graphs are labelled with A33 and A35. It is assumed from the methods that A33 is the vaccinia protein and A35 is the MPXV protein. However, this is worth clarifying throughout to avoid confusion as VACV, CPV and MPXV express different but similar named proteins A33 and A35.

Authors' answer: In this revised version of the manuscript, we have specified whether the antigen is derived from VACV, CPXV, or MPXV by labelling them accordingly, such as H3-VACV, A33-VACV, and A35-MPXV. This information is available in Table 1. We have kept the usual nomenclature in the figures, as we think mentioning the strain, from which the antigen was derived in the text is sufficient to avoid confusion and to keep the figures visually clean.

3. The paper referenced describes post-MPXV sera having preferential binding to the MPXV proteins A29, A35 and B6 over their VACV homologues. In contrast, post-MVA-BN sera showed preferential binding to VACV A27, A33 and B5 over their MPXV homologues. Importantly, both vaccination and infection produced antibodies to these proteins (less so for vaccination and MPXV A29 / VACV A27). However, it was suggested that these differences in binding specificities (i.e. the ratio of responses to homologues) may be used to differentiate between infection and vaccination.

Authors' answer: We have included an updated figure showing this differential binding response in a revised Figure 2. Here, we only focus on the antigen pairs, that show clear and significant differences, while antigen pairs with a less clear difference were included in a supporting figure (Figure S4). We have not specifically included those ratios in our analytical approach, but instead mentioned those differences as we think that such signals can be picked up by our ML-based approach.

4. Comparisons of VACV L1 and MPXV M1 were not done as suggested in the manuscript.

Authors' answer: This is correct and the main reason for that lies in the fact that, in our hands, we did not observe the same contribution of differential binding between L1-VACV and M1-MPXV (see Figure S4). This might have been caused by overall low responses to L1-VACV as well as batch-to-batch variability with L1-VACV, which was also the reason, why we have excluded this antigen pair from consideration for the ML training and testing.

5. Separately to this, the paper referenced showed that antibody responses to proteins MPXV A27 (N-terminal of A-type inclusion body protein) and MPXV M1 were significantly different in infected individuals versus those who had been vaccinated. Infected individuals produced a measurable antibody response to MPXV A27 whereas vaccination did not.

Authors' answer: We did not include MPXV-A27 as an antigen in our study, however we did find that the antibody response against the ATI-N-CPXV protein (N-terminal of A-type inclusion body protein) is significantly different in infected versus vaccinated individuals, with a measurable antibody response to ATI-N-CPXV only in infected individuals.

Line 191: *“In contrast, ATI-N-CPXV demonstrated a highly distinctive immune profile. IgG levels against ATI-N-CPXV were significantly elevated only in MPXV-infected individuals, distinguishing them from both pre-immune and MVA-immunized individuals, regardless of childhood smallpox vaccination status and panel, although the difference was less pronounced in the epi panel.”*

As the MPXV-A27 protein is the homologue to the CPXV N-terminal of A-type inclusion body protein, our results agree with this finding. As this point is important, but might be missed to the confusing nomenclature between VACV and MPXV, we emphasised this point by including the following statement in the discussion

Line 364: *“This aligns with prior work identifying MPXV protein A27—the homolog of ATI-N-CPXV—as a key discriminator of infection⁴², as well as binding to A35-MPXV/A33-VACV and B6-MPXV/B5-VACV homologue antigen pairs¹⁴. ATI has also been used to distinguish Dryvax from MVA immunization, as MVA-induced responses to ATI are absent due to a genomic truncation³¹. “*

6. In contrast, vaccinated individuals mount robust responses to MPXV M1 whereas infected individuals have weak/variable responses.

Authors' answer: In contrast to this study, we found that there were no measurable differences in the antibody responses of infected compared to vaccinated individuals to the M1-MPXV antigen. Although we did see some increase after MVA vaccination in contrast to MPXV infection or the pre-immune sera in the acute panel, this difference was small and only visible in subjects without previous smallpox childhood vaccination (see supporting Figure S2). The difference was not visible in the epi panel or in subjects after previous childhood vaccination. Hence, we did not further focus on this antigen, as we could neither confirm nor dispute this finding. In our hands, both L1 and M1 contributed only very little to differentiation. We have added a section on that topic to the discussion.

Line 307 *“In our study several highly immunogenic antigens including H3-VACV, E8-MPXV/D8-VACV, A35-MPXV/A33-VACV, and B6-MPXV/B5-VACV proved to be excellent markers for differentiating seropositive from seronegative samples^{32,47-51}, while others such as M1-MPXV/L1-VACV, despite still being immunogenic, exhibited lower reactivity and discriminatory power in our assay⁵².”*

7. It was this difference in response to MPXV A27 that was used to differentiate between infection and vaccination.

Authors' answer: Please see our answer to question 5 above, where we have added a discussion on that topic.

8. For accuracy, please reword to ‘it has been demonstrated that polyclonal antibodies produced post-infection or immunization exhibit differential binding to some of these antigens, namely A33 and B5. An observation suggested for serological differentiation by ELISA.’

Authors' answer: Thank you very much for this suggestion. During revision of our manuscript, we have shortened the introduction. Hence, this sentence was removed. However, we have cited the relevant publication and highlighted the contribution of A35 and B6 to the differential binding in the discussion.

Line 316: “...as well as binding to A35-MPXV/A33-VACV and B6-MPXV/B5-VACV homologue antigen pairs¹⁴.”

9. Creep of discussion into the results.

Authors' answer: We have thoroughly edited the manuscript and reorganized the information. We believe that all interpretations previously included in the Results section are now appropriately placed in the Discussion section. However, we have retained some interpretation in the Results section, as we believe it is necessary to guide the reader and clarify certain decisions related to downstream analysis.

10. Similar results were also demonstrated in <https://www.nature.com/articles/s41467-023-41587-x>

Authors' answer: We have included a discussion on that topic with relevant citations. Please see our answer to question 5.

11. The limitations of this work have not been discussed sufficiently.

Authors' answer: We have addressed the limitations of our work in the discussion.

Possible time-dependence of differentiation based on ATI-N:

Line 318: “*Misclassification of older individuals as MVA-vaccinated suggests that ATI-specific immunity from historic smallpox vaccination has largely waned, while responses to other orthopoxvirus antigens remain detectable for decades²⁰. This enables discrimination between MPXV infection, MVA vaccination, and residual smallpox immunity yet also indicates that ATI-N-CPXV may be more informative early post-infection, with decreasing sensitivity over time.*”

Caveats for training and testing of ML algorithms:

Line 325: “*As our data showed, training and testing machine learning models on mismatched serum panels (acute vs. epidemiological) reduced prediction accuracy, highlighting the importance of dataset selection to account for factors such as timing of infection, prior MVA immunization, and waning antibody responses. This diversity is essential for developing ML models with strong generalizability and reliable performance.*”

Confounders that negatively affect the assay performance:

Line 345: “*Despite these promising results, several limitations remain. The inherent variability in humoral immune responses can lead to misclassification, particularly among individuals with weak or waning antibody levels. Childhood smallpox immunization emerged as a relevant confounder, with many such individuals misclassified as MVA-vaccinated—a plausible outcome given that both exposures elicit similar immune profiles. From a seroepidemiological perspective, it is reassuring that misclassification as MPXV-infected was observed far less frequently. Notably, the machine learning (ML) models provided confidence scores for each prediction, enabling the identification of uncertain classifications. Nevertheless, further optimization is required to improve performance in borderline cases.*”

Responses to Reviewer #3

1. The novelty of the assay and ML analysis needs further clarification. Although the design of the multiplexed protein panel is impressive, and the approaches are well-conceived, similar methods have been applied to COVID-19 and other diseases. The authors should explicitly highlight their study's unique aspects and contributions compared to existing approaches.

Authors' answer: We have made changes to the text to highlight the differential and main contribution of our study. See discussion from lines 298 to 344.

Briefly, key strengths include:

- **High-resolution antigen panel:** Our assay includes immunodominant orthopoxvirus antigens (e.g., H3-VACV, E8-MPXV/D8-VACV, A35-MPXV/A33-VACV, B6-MPXV/B5-VACV), which enable robust discrimination between seropositive and seronegative individuals. ATI-N-CPXV reactivity was especially important for distinguishing MVA from MPXV.
- **Training data quality:** Cross-panel evaluation revealed that prediction accuracy is highly dependent on the composition of the training dataset. Using mismatched panels (e.g., acute vs. epidemiological) reduces performance, emphasizing the importance of epidemiological context.
- **Model performance:** Gradient Boosted Classification (GBC) outperformed Linear Discriminant Analysis (LDA), particularly in complex cases like MPXV breakthrough infections. GBC's ability to model non-linear immune patterns contributed to its superior accuracy and robustness, even without explicitly accounting for prior MVA or smallpox vaccination.

This approach highlights the utility of combining antigen-specific profiling with ML to disentangle overlapping serological signals and improve classification of orthopoxvirus exposure histories.

2. The manuscript demonstrates the effectiveness of the bead-based multi-antigen panel in distinguishing between mpox-infected and vaccinated individuals. It also shows the suitability of ML for handling the data. However, the manuscript lacks a detailed discussion on how ML specifically improves performance over traditional methods. The authors should include a more thorough explanation of how ML enhances assay performance and address any limitations that might be present in multiplexed assay results when analyzed without ML.

Authors' answer: We believe the key advantages of using machine learning (ML) include its ability to simultaneously analyse multiple antigens, its capacity for retraining with additional data to enhance predictive robustness, and its support for multiclass classification—unlike traditional assays, which are typically limited to binary classification. Furthermore, ML provides additional metrics, such as prediction confidence, which can strengthen the stringency of predictions. Lastly, we believe ML can complement traditional approaches, as both have their own merits. We have added data on the assay performance of traditional approaches using single antigens to our manuscript and supporting information (Figure S10, Tables S12 and S13):

Line 323: *“To compare our ML-based classification with binary classifiers utilizing single antigens, we performed receiver operating characteristic (ROC) analysis to establish robust population-based cutoff values. These thresholds enabled us to distinguish seropositive*

samples—arising from MPXV infection or MVA vaccination—from seronegative samples, as well as to differentiate MVA-vaccinated individuals from those infected with MPXV (Figure S7, Table S11). Among the single-antigen classifiers, ΔVACV, E8-MPXV, A35-MPXV, and B6-MPXV demonstrated strong performance in differentiating seronegative from seropositive samples, with F1 scores ranging from 0.87 to 0.94 (Table S12). Notably, ATI-N-CPXV emerged as the most effective single antigen for distinguishing MPXV-infected individuals from non-infected individuals (either pre-immune or MVA-vaccinated), achieving an F1 score of 0.83. While these single-antigen approaches yielded high discriminatory power, our ML-based method demonstrated superior performance, slightly outperforming ATI-N for identification of MPXV infections while enabling multiclass discrimination into MPXV, MVA and pre-immune.”

3. Given the complexity and non-linear nature of the results from the multiplexed panels, a deeper exploration of the data analysis techniques is needed. In such cases, neural networks might offer advantages over traditional ML algorithms in capturing complex, non-linear patterns. The authors should consider discussing why they chose ML algorithms over neural networks and whether they explored deep learning alternatives.

Authors’ answer: We have prepared a paragraph discussing our rationale for choosing traditional machine learning (ML) algorithms over deep learning approaches. Additionally, we have tested a deep learning approach to predict our classes, and data on its assay performance is included in the revised dataset provided with the manuscript. However, the deep learning approach performed worse than the methods selected in our study. We hope this clarification sufficiently addresses the reviewer’s concerns. Please let us know if you believe this paragraph is essential for understanding our work so that we can consider including it in the manuscript.

Line 213: “*A deep neural network (DeepTables) was also evaluated but excluded from primary analysis due to poor performance, likely reflecting the dataset’s small size, tabular structure, and lack of extensive hyperparameter tuning (see Supporting Data).*”

4. The study may not present the blinded sample set. While five-fold cross-validation is valid, it may not always reflect real-world, unseen data performance. The authors should include results from blinded sample testing to ensure robustness and confirm that the model generalizes well beyond the cross-validation.

Authors’ answer: We have fully addressed this point by measuring new samples to evaluate the performance of our trained models. The new data is presented in Figure 5 and discussed in the *Validation of the assay performance* section of the Results. Our findings indicate that assay performance was comparable to results obtained through five-fold cross-validation. We believe these additional measurements have significantly strengthened our manuscript by independently confirming the robustness of our approach. Moreover, the inclusion of challenging sera has helped identify areas for further assay improvement.

5. In the abstract and introduction, it would be helpful to briefly mention the number of samples tested with simplified information to provide the study's scale.

Authors’ answer: We added the numbers of sera to the abstract.

Line 32: “*Of six machine learning algorithms evaluated on sera from the 2022 outbreak and a follow-up cohort of at-risk men who have sex with men (MSM) six months later (n = 1260), the Gradient Boosting Classifier achieved the best performance (F1 = 0.82). Validation on an independent cohort (n = 143) maintained high classification accuracy, even in complex cases such as post-vaccination breakthrough infections.*”

6. The link for reference 12 is incorrect.

Authors’ answer: This is a URL link that ends with a parenthesis, a known bug in Endnote. We got in touch with their technical support and the alternative they offered us was to create a short link instead, which we did. We believe the problem has been solved.

7. In line 79, "two-fold" is unclear and should be used as another term.

Authors’ answer: This sentence was removed from the revised manuscript.

8. Figures 1b and S10 appear to be redundant. The IgM results in Figure 1b and the IgG results in Figure S10 should be removed to avoid duplication.

Authors’ answer: The figures have been streamlined to eliminate redundancies. We have significantly reduced the number of supporting figures and panels in the manuscript, ensuring a sharper focus on the core narrative of our study. We hope this revised presentation enhances clarity and conciseness.

9. Please check the labels in Figure S9, as there are some typographical errors. I also noticed similar errors in other figures, so a careful review is needed.

Authors’ answer: We carefully revised all figures and supporting figures in the manuscript.

10. In Figure 1a, consider using different colors for the lines. The current blue and green lines are hard to distinguish and too similar to the IgM/IgG color.

Authors’ answer: We have changed the colour for the fitted lines to grey (solid and dashed) to avoid confusion with the IgG and IgM colouring and to improve the discernibility between the blue and green lines.

11. Please switch Figures S14 and S15, as Figure S14 is mentioned after Figure S15 in the manuscript.

Authors’ answer: Figure S14 has been removed from the manuscript. Instead, the relevant information is included in the revised manuscript as parts of Figures 2 (panels e and f) as well as Figure S4.

12. Figures S18-S20 and S22-S23 should be replaced with higher-resolution versions. The labels in the plots are not clear.

Authors’ answer: Figures S18 to S20 and S22 to S23 were removed from the revised manuscript. Feature importance plots at higher resolution are provided as supporting data

instead, while the three-way ANOVA was omitted to further streamline the manuscript as requested by Reviewer #4 (see comments below).

13. The term "PBS/T buffer" in the Methods section (line 582) is unclear. It may refer to the incubation buffer, but this should be clarified.

Authors' answer: It refers to PBS/Tween 20. We have clarified this point. Additionally, we have moved the details for the methodology to the Supporting Information to streamline the manuscript.

Responses to Reviewer #4

1. The manuscript is very long and descriptive in nature. Too many unnecessary details consume the attention of what is actually important. Overall, this has led to a very bloated text. A revision should, therefore, include a substantial rewriting and narrowing of the key outcomes. This will decrease the risk of “data dumping”.

Authors’ answer: We have significantly revised the text by removing unnecessary information and reorganizing the flow. We believe it is now streamlined and more effective. However, we have also added new data, as requested with regards to validation measurements and comparison with classical analysis using binary ROC classification (only briefly in the manuscript and mostly included as supporting information). Hence, we hope, that we found the right balance between shortening the manuscript, focussing on key outcomes, and adding the requested new measurements and analysis.

2. Analyzing the data in all directions is an interesting exercise, but it does not increase the value of the outcome. A clear strategy and clean study design need to be defined and explained before building a useful classifier. Avoid any side stories that could distract from the main focus. To us readers, the work appeared as if they had simply included all results from their data analysis. This is not impressive but distracting. We expect from authors of Nature Communication submissions the ability to moderate and guide readers, rather than leaving them with the data. We also noted that single results were sometimes linked to several figures and tables (both main and supplementary). Jumping between different sub-studies makes it difficult to find what the significance of this work actually is.

Authors’ answer: We have significantly revised the text by removing unnecessary information and reorganizing the flow. We believe it is now streamlined and more effective. Specifically, we have removed the three-way ANOVA and instead focussed only on relevant and exemplary antigens (new Figure 2, full results in the supporting information). Additionally, we have reduced the number of figures, tables, and panels, concentrating on the key algorithms and results. We hope these revisions enhance the clarity of our main message.

3. The multi-parameter assay concept as such is neither novel nor has their model and approach been proven to be informative in independent data sets. The recent pandemic has inspired the community to develop several similar approaches and concepts. A new literature survey will reveal such efforts.

Authors’ answer: Whilst there is a large body of publications where the development and use of multi-parameter immunoassays have been tested, there are few multi-parameter assays that use ML guided analysis to evaluate the complexities of the antibody response. We believe that the publications such as ours that evaluate different ML algorithms are valuable to help advance this fast-growing field of ML guided analysis. The use of both epidemiological and acute serum panels for assay development is also not common during multi-parameter assay development and further enhances the value of our work. Finally, we included an independent validation on new measurements on the revised manuscript and believe, this validation further strengthens our assay. We have also mentioned other studies using ML and multiplexing serology in the discussion:

Line 298: *“A significant aspect of our assay, compared to the previous studies distinguishing between MPXV infection and immunisation responses, is the use of machine learning algorithms for data analysis. While the extensive data gathered by multiplex assays make the*

use of machine learning algorithms a logical choice for innovative data analysis, there are still relatively few studies that have employed this approach for differentiating between various serum statuses. Previous studies applied machine learning to serological multiplex data to distinguish recent from historical malaria transmission^{59,60}, to improve serodiagnosis for SARS-CoV-2 (Severe Acute Respiratory Syndrome Coronavirus 2)⁶¹ or to identify cases of active tuberculosis⁶².

4. Differentiation of infection and vaccination depends on several factors, including the differences between vaccine and viruses, time since infection and vaccination as well as unrelated events interfering with the readout. This must be investigated.

Authors' answer: We agree that several factors contribute to the possibility to differentiate between infection and vaccination. To emphasise this important point, we have added the following paragraph:

Line 345: *“Despite these promising results, several limitations remain. The inherent variability in humoral immune responses can lead to misclassification, particularly among individuals with weak or waning antibody levels. Childhood smallpox immunization emerged as a relevant confounder, with many such individuals misclassified as MVA-vaccinated—a plausible outcome given that both exposures elicit similar immune profiles. From a seroepidemiological perspective, it is reassuring that misclassification as MPXV-infected was observed far less frequently.”*

We believe we have thoroughly investigated multiple factors that could influence differentiation by:

- i) testing our algorithm on both an acute and an epidemiological panel,
- ii) performing independent validation measurements,
- iii) analyzing and demonstrating the dependence of the immune response on childhood vaccination status,
- iv) assessing the impact of breakthrough infections,
- v) identifying ATI-N as a key differentiator between the MVA vaccine and the MPXV virus, and

We hope this sufficiently addresses the concern. If further clarification is needed, we would appreciate the reviewer specifying how we can better address these issues.

5. It remained unclear to us if different antigen panels were used for the different study sets.

Authors' answer: The same antigen panel was used throughout the study and has been clarified in line 102 *“This 19-plex assay was applied to all serum panels described in this study.”*

6. Please explain better why retrospective differentiation of mpox vaccination from infections is of value. Add a demographic table of the two study sets into the main text.

Authors' answer: This has been explained better in the introduction:

Line 60: *“Serological assays capable of distinguishing infection-induced from vaccine-induced antibodies are critical to estimate true infection rates, detect unreported or asymptomatic*

*cases, and evaluate the effectiveness of vaccination programs. Such tools also support studies of infection dynamics in at-risk populations*³. “

Additionally, we have added a table containing the demographic information about the serum panels used in this study as Table 2.

7. Line 104ff discusses the classification performance, but the authors do not explain why. Is it the sensitivity of the assay, waning of Ab levels, immunocompromised donors, or distribution of the data?

Authors' answer: We have moved the discussion of the classification performance to the discussion, where we elaborate on the difference with regards to the panels tested.

Line 325: *“As our data showed, training and testing machine learning models on mismatched serum panels (acute vs. epidemiological) reduced prediction accuracy, highlighting the importance of dataset selection to account for factors such as timing of infection, prior MVA immunization, and waning antibody responses. This diversity is essential for developing ML models with strong generalizability and reliable performance.”*

8. Line 151ff: explain “complex Delta antigen”.

Authors' answer: We have added further explanation on this antigen to the manuscript.

Line 96: *“To capture the complexity of the antibody response not covered by the recombinant proteins, we also incorporated lysates from VACV-infected and uninfected cells (included as a negative control). The antibody response to VACV-infected cell lysate with binding to uninfected cell lysate subtracted is hereafter referred to as Delta-VACV^{17,32}.”*

9. Line 152 on Ab levels: It is important to remember that the reported MFI values are a combination of factors and not only Ab levels. The authors mention polyclonality of the immune response but should add that this also creates Abs of different affinities. MFI is not only abundance.

Authors' answer: We agree with the reviewer's objection. We used the term “antibody levels” instead of MFI, because we did not report MFI values but instead quantified antibody levels using VIG as standard. Hence, using MFI would be incorrect. As we saw, that this point was not made sufficiently clear already at this part of the manuscript, we have added the following clarification.

Line 113: *“To further assess the immunogenicity of the selected recombinant antigens and their contribution to the overall antibody response detected in the reference assays, we determined correlation coefficients between individual recombinant antigens and the complex Delta-VACV antigen in our multiplex assay with the three reference assays tested. The strongest immune responses, characterized by a broad dynamic range (indicative of both high antibody affinity and abundance) were directed against Delta-VACV, D8-VACV/E8-MPXV, H3-VACV, A33-VACV/A35-MPXV, and B5-VACV/B6-MPXV (Fig. S1)”*

10. Line 155ff: Using other related correlation methods does not provide independent evidence. It's a circular argument.

Authors' answer: We think this might be a misunderstanding. We have used either Pearson's r or Spearman rank correlation, where methodically fitting. Methods containing titres (IFA, NT) were correlated using Spearman rank correlation while ELISA were correlated using Pearson's r correlation coefficients. After streamlining our manuscript, we have included this information in the legend to Figure 1.

Line 716: "*b) Pearson correlation of multiplex IgG binding to individual recombinant antigens with ELISA results. c, d) Spearman correlation of IgG responses to recombinant antigens in the multiplex assay with IFA titers (c) and NT titers (d)*"

11. Line 160ff: The authors have no evidence about the immunogenicity, as their observations could be driven by technical biases that favor one antigen over the other. Neutralization assays or other means of normalizing between antigens must be added to support such claims.

Authors' answer: We agree with the reviewer, that we did not test immunogenicity of single antigens directly, but instead measured the polyclonal immune response against those antigens indirectly. Normalizing to neutralization assays, however, could also be misleading as antigens targeting some antigens (e.g. L1) contribute strongly to orthopoxviral neutralization whereas antibodies targeting other antigens (e.g. A27) contribute less or in a cell-type dependent mode of action, and antibodies against other targets are not neutralizing, at all (see, among others Stern, D. *et al.* Development of a Genus-Specific Antigen Capture ELISA for Orthopoxviruses - Target Selection and Optimized Screening. *PLoS One* 11, e0150110, doi:10.1371/journal.pone.0150110 (2016)). We did, however, normalize our results to exclude technical biases, to a standard dilution of VIG, which was included on each plate that was measured, and the MFI reading against each antigen in every serum sample was normalised and quantified against this VIG standard. This is described in the manuscript.

Line 454: "*To normalize binding responses across plates, a standard curve of five VIG dilutions was included on each plate. VIG IgG concentrations ranged from 10,000 $\mu\text{g}/\text{mL}$ to 39 $\mu\text{g}/\text{mL}$ in five 1:4 serial dilutions. For each orthopoxvirus-specific antigen, a 4-parameter logistic regression model was fitted to \log_{10} -transformed mean fluorescence intensity (MFI) versus \log_{10} IgG concentration using the drLumi package⁶⁰, with buffer-only controls used to constrain the curve's lower bound. Serum dilutions (1:100 and 1:1000) were interpolated against the standard curve per antigen.*"

12. Line 179: Explain how participants "self-reported mpox infection".

Authors' answer: We have added an explanation in the manuscript.

Line 139: "*Classification of sera was based on self-reported MPXV infection ($n = 59$), MVA vaccination ($n = 476$), or neither ($n = 324$), as indicated in a questionnaire completed by participants in a sero-epidemiological study³³*"

13. Line 185ff: The manuscript in its current form reports on applying different ML methods, but it does not justify their selection over other models and does not benchmark them against less complex statistics.

Authors' answer: We have justified our selection of GBC and LDA over other tested algorithms, as GBC demonstrated the best overall performance, while LDA was more robust and specific in handling misclassifications in the pre-immune serogroup (see Fig. 4). We

have also discussed the limitations of the LDA algorithm in classifying challenging breakthrough infections. Additionally, we benchmarked the differentiation between infected and non-infected based on ATI-N using ROC analysis against binary classification using GBC on the validation panel. We have also discussed the use of single antigens as an alternative in resource-limited settings.

Line 278: *“To explore simplified classification options, we compared ML-based predictions to threshold-based single-antigen classifiers using ROC analyses to distinguish MPXV infected individuals from others in the validation serum panel (Fig. S10, Table S12). Among all antigens, ATI-N-CPXV performed best in distinguishing MPXV-infected individuals, achieving an F1 score of 0.83. Binary classification using GBC achieved slightly better results, with an F1 score of 0.85 and higher specificity (0.95 vs. 0.91 for ATI-N-CPXV alone; Table S13). These findings highlight the added value of multiplexed, ML-based analysis—particularly in complex immunological backgrounds—while supporting ATI-N-CPXV as a strong single-marker candidate when resources are limited.”*

Line 334: *“Our comparative analysis of LDA and GBC highlights key algorithmic trade-offs relevant for serological classification. The lower performance of LDA in distinguishing MPXV breakthrough infections from MVA likely stems from its assumptions of Gaussian-distributed features and a shared covariance matrix, which may not accurately reflect the antigen distribution. If antigen responses are non-normally distributed or exhibit complex interactions, LDA’s linear boundaries struggle to separate the classes. In contrast, GBC can model non-linear relationships and is more robust to outliers, making it better suited for capturing subtle immune response patterns that LDA may overlook⁵³. Remarkably, GBC achieved this level of performance without explicit modelling of prior MVA vaccination before MPXV infection or childhood smallpox immunization during training. When evaluated on an independent dataset, GBC alone demonstrated strong predictive power in the detection of breakthrough infections.”*

14. Line 196: Explain “presumed childhood infection”.

Authors’ answer: We could not find any instance of “presumed childhood infection” in our text. What we found was presumed childhood vaccination or immunisation, which refers to a population of individuals over 50 years old who most likely received smallpox vaccine in childhood. Please clarify whether this is what you are referring to. If yes, we hope the following paragraph clarifies the reviewer’s question:

Line 149: *“Additionally, we applied an age-based classification to assess the impact of presumed childhood smallpox vaccination as a potential confounder. Individuals older than 50 years as of 2022 (year of birth < 1972) were classified as likely to have received smallpox vaccination in their early childhood due to mandatory vaccinations at that time, those under 40 as unlikely, and those between 40 and 50 as ambiguous (excluded from age stratified analysis).”*

15. Line 205: Explain “simple main effects”.

Authors’ answer: The “simple main effects” we analysed were the contributions of single antigens on the classifications into the three statuses with regards to infection or immunisation (pre, MVA, MPXV), depending on the childhood immunisation status (Yes or No). This analysis was performed, as the three-way ANOVA showed significant interactions between

antigen, childhood immunisation status, and serostatus with regards to infection or immunisation. Our intention here was to inform readers that we investigated the individual effects of each antigen on the IgG response. As one point of criticism was the inclusion of too many data points, we have significantly shortened the manuscript with regards to the three-way ANOVA and instead focused on significant differences between single antigens, that contribute significantly to the classification of sera.

16. Line 207ff: It is confusing to read about different selections of antigens (= panels) and sample sets. Did they use different sets of beads? Did they analyze the samples together? Was the analysis done blinded? This needs to be validated in a new sample set from another collection.

Authors' answer: There is only one antigen panel, with each antigen always being bound to the same bead region. This was clarified under comment #5.

With regard to the second question (“Did they analyse the samples together”): Different sample sets were measured consecutively, starting with the sera from the “acute” dataset to set up the assay initially, followed by sera from the “epi” dataset, which was measured after setting up the assay. The samples from the “acute” dataset were not measured blinded while the samples from the “epi” dataset were measured blinded and unblinded for analysis, which was needed to train the supervised ML-algorithms that we have trained. We have clarified this point in the manuscript.

Line 396: *“Sera from the acute panel were used to develop the assay and were measured unblinded. The epi panel was processed subsequently in a blinded manner using three bead lots, then unblinded for ML model training. This minimized systematic bias for the epi panel, though not for the acute panel.”*

Finally, we have validated our assay on independent measurements (see new results under “Validation of the assay performance” lines 255 ff)

17. Line 211ff: Explain why the IgM levels were lower.

Authors' answer: In the revised manuscript, we have focused solely on the overall IgM response. Therefore, we did not further emphasize differences related to childhood vaccination or naïve subjects. However, the lower IgM levels can be explained by the nature of the immune response: in naïve individuals, the response is initially IgM-driven, whereas in those with childhood vaccination, it shifts toward IgG due to the presence of immunological memory.

18. Line 219ff: Explain the sentence “However, differentiation between... “. It does not make much sense as is.

Authors' answer (line 182): This sentence was replaced with: *“However, due to substantial overlap between the MVA and MPXV serogroups, these antigens alone were insufficient to reliably differentiate MPXV infection from MVA immunization.”*

19. Line 259ff: The statement about batch effects raises concerns about the overall quality of the data (precision, reproducibility, reliability,...). Add a PCA or UMAP overview, plus a metrics table (Excel sheet) to summarize the assay performance and batch variability. Explain

how the randomization across analysis batches was handled to avoid any upfront bias that cannot be accounted for downstream.

Authors' answer: We understand the concerns of the reviewer, but our data show that our quality control measures were effective due to the following reasons. First, due to our QC measures (control coupling efficacy for each batch of reagents), we were able spot deviations in the coupling efficacy (see also Figure S5). Out of all antigens, that were included in our assay, we observed this only for one antigen (L1), which was consequently excluded from further validation. Furthermore, L1 and the MPXV homologue M1 were among the antigens that showed the least reactivity when tested by different sera as well as the VIG control (together with ATI-C and A5L), implying that its exclusion had a low impact on the overall classification performance.

A complete randomization of samples over different reagent batches was not possible, as the first models were trained and tested on the acute panel. Those data lead to the decision to use our assay on the epidemiological panel. Here, as we have mentioned earlier, measurements were performed blinded across three different bead batches and more than 50 96-well assay plates. Hereby, systematic bias introduced by different reagent batches can be excluded for the measurements on the epi panel, but not completely ruled out for the acute panel.

To be fully transparent about this limitation, we included the following statement in the appropriate section of the materials and methods part of the revised manuscript.

Line 396: *“Sera from the acute panel were used to develop the assay and were measured unblinded. The epi panel was processed subsequently in a blinded manner using three bead lots, then unblinded for ML model training. This minimized systematic bias for the epi panel, though not for the acute panel.”*

We have prepared a paragraph on quality control measures, that could be included, if needed. We omitted it, so far, to keep the discussion short.

“To enhance the overall reproducibility of our results and the robustness of our assay, we incorporated standard curves of vaccinia immune globulin (VIG) on each measured plate. The inclusion of internal references significantly minimized plate-to-plate and batch-to-batch variability, ensuring consistency across experiments⁵⁷. Moreover, VIG served as a critical control for batch-to-batch variability by enabling direct comparison of antigen reactivity between older and newer batches, thereby maintaining assay performance across different production runs. This stringent quality control framework allowed us to identify and exclude antigens with suboptimal performance, such as LI-VACV, from further analysis. Additionally, the inclusion of human serum albumin (HSA) as a negative control could provide helpful in mitigating background binding, which can occur sporadically in individual sera and may otherwise lead to false-positive results. Such artefacts remain undetected when relying solely on median fluorescence intensity (MFI) raw data to establish cut-off values, as it is usually done⁴⁸⁻⁵⁰. Without these quality control measures however, both false-positive and false-negative rates could increase, compromising assay accuracy and reliability. Finally, by applying both our ML algorithms as well as cut-off values determined by ROC analysis on novel validation measurements, we could show, that both methods are robust and transferable on independent measurements.”

Finally, unlike other published assays, we have tested different batches with each batch being qualified by comparison between old and new batches. Additionally, we measured reproducibility of our results with independent experimenters and batches (e.g. see Fig 1 below).

Fig. 1. Reproducibility of measurements tested in two independent measurement on the validation panel for IgG and IgM measurements. The measurements were performed by two operators using two batches of coupled beads on two different instruments.

20. The performance of data models is highly dependent on what the classifier was trained on. The less prototypical and accurate that data is, the less selective and precise its application will be. Interchanging and reusing data for such purposes is not a sustainable practice. It must be much clearer how the authors used their data for such proposes and any reuse of data is discouraged. The authors need to decide on one approach and not flip their model around between the different data sets to find a story worth telling. See also line 291 and explain what they mean by “homologue panel”.

Authors’ answer: The use of the term "homologue" to refer to the panels was confusing, as we used the same term for the antigen pairs. To clarify the text, especially when referring to a panel, we have adopted the terms "matching panels" to indicate when the same panel is used for both training and testing, and "mismatched panels" to denote instances where not the same panel was used for training and testing the algorithms. With regards to the impact of the performance of the ML models on the data that was used to train the classifier, we completely agree with the reviewer. In fact, we made the comparison between different models trained on matching or mismatched panels specifically to emphasise this point. This is, in our view, not reuse of data but instead the evaluation of different algorithms as well as the impact of the immunological signature of the mismatched serum panels on the outcome and performance of the ML algorithms. We have tried to make this point clear in the following paragraph, which has been amended to include the criticism raised by the reviewer.

Line 324: “*Second, by evaluating several different ML algorithms across epidemiologically diverse serum panels, we highlight the importance of robust training data. As our data showed, training and testing machine learning models on mismatched serum panels (acute vs. epidemiological) reduced prediction accuracy, highlighting the importance of dataset selection to account for factors such as timing of infection, prior MVA immunization, and waning antibody responses. This diversity is essential for developing ML models with strong generalizability and reliable performance.*”

To further streamline our analysis and the manuscript and to take up the criticism, we have focused our deeper analysis of the assay performance on the two best performing and most robust algorithms (GBC and LDA).

21. Line 273: Please explain why they “did not further stratify...”.

Authors’ answer: We performed initial experiments in which we further stratified the classification task by presumed childhood immunization status, inferred from the year of birth. This analysis showed no significant improvement in assay performance across strata (data not shown). One possible explanation is that increasing the level of granularity reduced the number of sera in each stratum, which in turn decreased the size of both the training and test datasets. Consequently, the algorithms had fewer samples on which to be trained and tested. In addition, birth year information was missing for many sera, particularly in the epidemiological (epi) panel, preventing these samples from being used in algorithm development. We have added the above explanation to the manuscript.

Line 208: “*Although some immune responses were likely influenced by childhood smallpox vaccination, we did not stratify by age due to incomplete metadata across the panels.*”

22. Line 274: Explain which data in “...did not have sufficient data on...” they refer to.

Authors’ answer: Please see our answer to question 21 above. The necessary data on year of birth was not available for enough sera.

23. Line 280: Explain why “5-fold” and not 100-fold.’

Authors’ answer: A 5-fold cross-validation approach has been shown to be sufficient for robust model validation. A larger number of folds (e.g., 100-fold CV) reduces bias, but increases variance in the performance estimate, as each training set is smaller. On the other hand, 5-fold CV strikes a good balance between bias (overestimating model performance) and variance (the variability in performance across different folds). Research has shown that 5-fold and 10-fold cross-validation generally provide reliable performance estimates that are robust enough for most machine learning tasks. In many real-world scenarios, the extra computational effort of 100-fold cross-validation does not add much value in terms of generalization performance. A 5-fold CV is typically sufficient to assess a model’s ability to generalize and ensure that results are not overly sensitive to the particular choice of training and testing splits.

24. Line 329ff: Explain the sentence “Here we found...classifications...were increased...”.

Authors’ answer: The entire paragraph was revised and we believe that it is clearer. It reads as follows:

Line 239 “Based on overall performance, we focused further analyses on GBC and LDA—two of the most robust classifiers trained on the combined dataset (“all”). To assess the potential confounding effect of childhood smallpox vaccination, we stratified misclassification rates by age as a proxy for vaccination status as before.

As shown in Fig. 4b, among presumed smallpox vaccinated individuals, ~30% of pre-immune sera were misclassified as MVA by both models, while 14% (GBC) and 7% (LDA) were misclassified as MPXV. In contrast, younger, presumed non-vaccinated individuals showed notably lower misclassification rates of ~ 2%. These findings suggest that age-associated residual immunity contributes to orthopoxvirus-specific antibody profiles, complicating accurate classification.”

25. Line 338ff: The statement “...excluding low confidence predictions, which also have a higher rate of misclassifications...” is another example of circular reasoning.

Authors’ answer: We do think that the prediction confidence is a valuable metric that is able to define an area of uncertainty with regards to the classification of sera. Defining such an area of uncertainty is common in serology and is usually defined around the cut off values (equivocal or ambiguous results). The immune profile with regards to the classification of sera as either MPXV infected, MVA immunised, or pre-immune could also lie within such an area of uncertainty due to several reasons: too short a duration after an infection before sampling, where there was insufficient time to mount a proper immune response, previous MVA immunisation before an infection or vice versa, overall low immune response, lowered antibody levels after extended duration following infection or immunization.

The decision to exclude low-confidence predictions is not driven by a direct improvement in overall accuracy per se, but rather by the goal of ensuring higher confidence in the remaining predictions. In some cases, particularly when the aim is to minimize uncertainty and increase the robustness of the results, it can be beneficial to exclude predictions that fall below a certain confidence threshold. This allows for a more reliable classification of the remaining samples, where higher confidence is prioritized over the inclusion of potentially ambiguous or borderline predictions.

In our revised manuscript, we have applied this metric conservatively, excluding only predictions with a confidence score of 0.5 or lower. However, we consistently report metrics for both scenarios: one excluding low-confidence predictions, and one including the entire dataset.

26. Line 340 is one of the first to introduce the term “cut-off value,” but there is no explanation of what that cut-off is and how others could make use of this for clinical decision-making. It raises the question of how the MFI data was treated and if it was assessed for the criteria needed to fulfill the modeling requirements. Would similar outcomes be reported if the MFI were to be converted into binary classifiers?

Authors’ answer: The cut-off value in line 340 defines the classification probability, which optimises the trade-off between exclusion of samples, for which no confident prediction can be made, from samples, with a higher probability of being correct. In the revised manuscript, we have used the classification probability of the ensemble learning approach and LDA with a cut-off of above 0.5.

As this threshold does not apply to MFI values, we are not sure about the other questions from the reviewer and would kindly ask for further clarification of the question, if needed.

27. In essence, the paper is based on machine learning, so one can expect some complexity in the data analysis approach and data representation. However, it is somehow difficult to understand which is the final panel of antigens and why it performs best in classifying certain phenotypes. Despite its timing, it is confusing for us that IgG, IgM, and IgG+IgM data are discussed interchangeably in different contexts. Whatever model is the most suitable, there needs to be an explanation for the observations.

Authors' answer: We have re-written the manuscript and believe that it is now clearer that only one antigen panel was used to test all the samples (see comment 5) and why IgM data was also included in the analysis.

For the further display figures and text, we have focussed on IgG to reduce redundancy and to streamline the manuscript. We have included IgM results in the Supporting Information. We have added explanations for the better suitability of GBC over LDA in the discussion.

Line 334: *“Our comparative analysis of LDA and GBC highlights key algorithmic trade-offs relevant for serological classification. The lower performance of LDA in distinguishing MPXV breakthrough infections from MVA likely stems from its assumptions of Gaussian-distributed features and a shared covariance matrix, which may not accurately reflect the antigen distribution. If antigen responses are non-normally distributed or exhibit complex interactions, LDA’s linear boundaries struggle to separate the classes. In contrast, GBC can model non-linear relationships and is more robust to outliers, making it better suited for capturing subtle immune response patterns that LDA may overlook⁵³. Remarkably, GBC achieved this level of performance without explicit modelling of prior MVA vaccination before MPXV infection or childhood smallpox immunization during training. When evaluated on an independent dataset, GBC alone demonstrated strong predictive power in the detection of breakthrough infections.”*

28. There is too little discussion about the clinical implications and interference for other events between vaccination/infection and sampling.

Authors' answer: Our assay aims mostly at being a tool for epidemiological studies. This has been clarified in the discussion.

Line 365: *“In conclusion, our novel assay integrates the power of multiplex serology and machine learning to achieve differentiation between MPXV infection and MVA vaccination. With its high-throughput capability and minimal sample processing, this method offers significant advantages for public health surveillance, facilitating efficient monitoring of MPXV infections, informing vaccine strategies and outbreak responses against emerging viral threats.”*

Hence, we don't see that our assay has any clinical implications, as acute diagnostics is performed by PCR. However, we are unsure if we understood the question correctly. Which events between vaccination/infection and sampling should be discussed? We kindly ask for clarification, so we can take up this point in our manuscript.

29. The authors should check abbreviations/acronyms throughout the manuscript because some are not cited. e.g. IMV and EEV in Tab 1

Authors' answer: We reviewed the entire document, including figures and tables, and defined all abbreviations and acronyms, leaving out just very well-known terms such as PCR, PBS, BSA.

30. Terms like good, better, higher, and largely ... are imprecise and must be defined.

Authors' answer: The entire text has been edited and we checked every instance of these words and double checked that they would be followed by either an explanation, or linked to a table or figure where more information could be found, or a reference.

31. The figures include many panels, and the axis, and p-values are difficult to read (e.g., Fig 3). The authors may consider reducing the panels and selecting the most interesting plots.

Authors' answer: We have taken up the reviewer's suggestion, also in conjunction with our response to question 20, and have focussed our analysis on the most important aspects. We have reduced the number of panels and analysis in our revised manuscript and tried to shorten our paper, despite the inclusion of novel measurements. We hope the new version is clearer, more streamlined and easier to understand.

32. Fig 1. Which samples were used to correlate the multiplex bead assay with the IFA and NT assays? Fig 1a. How were the intervals in x-axis selected for plots 3-6?

Authors' answer: A detailed explanation of which samples were used to correlate the multiplex bead assay with the IFA and NT assays is provided in the online methods, section "Serum panels and ethics statements".

Line 377: *"To establish and benchmark the multiplex assay, we used a comparison panel comprising sera from PCR-confirmed CPXV or MPXV infections and from MVA vaccines. Sera were selected to cover a broad range of IgG and IgM reactivities. For IFA comparison, 25 sera were tested (20 CPXV, 4 MVA, 1 Parvovirus B19). ELISA benchmarking included 14 sera from 5 individuals at pre-, post-prime, and post-boost stages of MVA vaccination. We also included 36 sera from 18 PCR-confirmed MPXV cases (≤ 33 days post-infection), and 21 sera previously tested by IFA for ELISA cross-validation. NT comparison involved 25 sera from 10 MPXV cases tested for VACV-neutralization as previously described¹⁷."*

The intervals in the x-axis for plots 3 and 4 represent the dilutions of the serum samples that were used in the IFA. The intervals in the x-axis for plots 3 and 4 are NT titres, that were quantified as described in reference 17.

33. Fig 1b and line 152: the authors refer to "highest autoantibody levels" detected for some of the tested antigens, but it seems that all antigens reach similar levels in Fig 1b.

Authors' answer: We can see, that this finding is not obvious due to the way the results have been described, so we tried to add an explanation about how the data has been quantified using VIG as standard curve. To clarify the different starting points of the quantified values, we have added a short explanation.

Line 467 *"For antigens, against which VIG is weakly reactive, quantified values are hereby higher, as higher concentrations of VIG are needed to reach certain MFI levels in comparison to antigens, against which VIG is highly reactive. Hence, the overall dynamic range is indicative of antigenicity, whereas the absolute values represent the reactivity of VIG against the antigens."*

It is important to note, that these biases have, of course, been accounted for before the ML analysis by normalisation of all data as reported in our manuscript at line 478: *"Input data*

consisted of normalized (min–max scaled) IgG and IgM measurements from the acute and epidemiological (epi) serum panels.”

34. Fig 2 represents that the acute and epidemiological sample panels were tested for either IgG or IgG+IgM, but it seems to me that the results report either IgG OR IgM (see Fig 3)

Authors’ answer: This is correct. We have not trained our ML models on IgM data alone, as IgM alone would not be sufficient in the epi panel. In our revised manuscript, we only show IgG data to reduce the number of panels shown in the figure. As we see that this might lead to confusion, we have adjusted Fig 2 (now Fig 3) to read “IgG and IgM” to make it clear, that results were not mathematically added but instead both were considered together.

Point-by-Point Reply

We thank all reviewers and highly appreciate their time, effort, and thoughtful critiques, which have helped strengthen our manuscript. We have addressed all remaining points in full. Although the underlying data are unchanged from our first revision, we added targeted analyses to clarify and better communicate the key advantages and conceptual advances of applying ML to multiplex serology compared with traditional approaches. We believe these revisions resolve the issues raised by Reviewer #4 while retaining the structure and content that Reviewers 1–3 considered suitable for publication.

To keep the manuscript succinct despite the new analyses, we streamlined the presentation: we reordered figures, moved the former Figure 1 to the Supplementary Information, and focused the main text on essential results. We also revised Figure 2 (ML workflow) to clarify terminology, the overall experimental design, and the use of distinct serological cohorts for training, testing, and independent validation.

For a complete comparison between Revision 1 and Revision 2, please consult the tracked-changes version of the manuscript. Key changes made in direct response to Reviewer 4's comments are highlighted in the point-by-point reply, with all line numbers referring to the tracked-changes document.

We trust that these clarifications further strengthen our key message while keeping all relevant data in place. We are grateful for the constructive feedback, which helped us clarify our message for a broad readership.

Best regards,

Daniel Stern (on behalf of all authors)

REVIEWER COMMENTS

Reviewer #1 (Remarks to the Author):

All previous concerns have been addressed in the revised manuscript. Thank you.

Reviewer #2 (Remarks to the Author):

Reviewer #3 (Remarks to the Author):

Thank you for the thorough and thoughtful revisions. I am satisfied with the changes made in response to the comments. The manuscript is significantly improved in clarity, robustness, and presentation. I support the acceptance of the revised version.

We sincerely thank Reviewers 1–3 for their time and effort in evaluating our manuscript and their positive assessments of our revision. The additional revisions made in response to Reviewer 4 are intended to address the remaining points while preserving the changes already endorsed by Reviewers 1–3. We hope these updates further improve the clarity and robustness of the manuscript.

Reviewer #4 (Remarks to the Author):

- 1) We thank the authors for their substantial efforts to improve the manuscript in terms of length, structure, and figures. The overall story has become somewhat clearer, and the flow of the results is easier to follow.*

We sincerely thank the reviewer for acknowledging the improvements in the manuscript's structure, clarity, and flow. We really appreciate the constructive feedback and guidance that helped us refine the presentation, and we are glad that the revised version reads more clearly and cohesively.

- 2) However, and despite this major revision, our level of excitement about the insights derived from the work has not increased significantly. We do not have any stakes in this area of research, but as readers and experts in proteomic studies of the immune system, the intermixed messaging about ML/AI and multiplexed serology left us with the question about the impact of the work and how it would change the community's approach to adapt to their observations.*

We sincerely thank the reviewer for their continued engagement with our manuscript and for sharing their perspective. We understand the concern regarding the clarity of our central message, particularly regarding the interplay between ML/AI and multiplex serology and the implications for the broader field.

To address this, we have revised the manuscript to better articulate the conceptual advance enabled by the integration of these methods. Specifically:

We now more explicitly describe the unique role of multiplex serology in capturing high-dimensional immune signatures that are not discernible through traditional single-antigen or threshold-based assays.

To underscore this point, we have included additional analyses directly comparing the GBC algorithm with the two best-performing single antigens, ATI-N-CPXV and E8-MPXV, but also the second-best performing antigens (B6-MPXV, ratio of A35-MPXV/A33-VACV) across both the combined (acute and epi) and validation cohorts (see our responses to comments 8 and 9). In every scenario examined, the GBC model outperformed not only each single antigen individually, but also the combined use of both antigens, as suggested by the reviewer.

More importantly, in an additional benchmark study, our results clearly demonstrate that the multiplex approach combined with ML-based analysis consistently outperforms single-antigen methods, even when the most influential antigens (ATI-N-CPXV and E8-MPXV) are excluded from the panel. Notably, the ML model maintained high classification performance after the removal of the two most powerful single antigens, highlighting the robustness and resilience of this approach. Conversely, when ATI-N-CPXV was removed and discrimination relied on the A35-MPXV/A33-VACV ratio in a classical threshold-based approach, performance dropped significantly.

These findings are particularly relevant for research questions where no single antigen is sufficiently specific, such as distinguishing immune responses to closely related virus strains. Our results show that considering several homologous antigens within a multiplexed assay, analyzed using ML, is both sufficient and robust for differentiating even highly cross-reactive antibody responses. This underscores the unique ability of ML-based multiplex serology to reveal complex immunological patterns that remain undetectable using conventional single-antigen approaches.

Finally, we now frame the broader impact more clearly in the Discussion: our work offers a generalizable framework for applying ML-enhanced multiplex serology to other pathogen systems with cross-reactive antigen profiles (e.g., dengue/Zika, hantaviruses), and suggests a shift toward integrative serological analytics in public health surveillance.

However, we would like to emphasize that our approach is not universally superior to classical methods of serological assay analysis. As the reviewers rightly point out, both the training/testing process and multiplexing can be labor-intensive and time-consuming. Therefore, the potential benefits of our method must be carefully weighed against the resources required to establish and maintain such an assay.

We fully acknowledge that in resource-limited settings or for more straightforward targets or classification tasks, simpler approaches are often sufficient. Nonetheless, we can show that our method offers a distinct advantage in differentiating antibody profiles induced by, for example, closely related virus strains or antigens. In such cases, classical methods often struggle due to subtle, variable, or overlapping differences in responses to individual antigens. In contrast, our approach leverages patterns across a broader antigen panel, providing robust and reproducible discrimination.

Revised sentence (line 537):

“The key strength of our approach is its ability to effectively dissect the cumulative effect of individually weak and overlapping patterns in the immune response, where single antigens fail to distinguish between groups.”

Revised sentence (line 545):

“Our work thus introduces an integrated framework that combines multiplex serology with machine learning. By resolving immune responses to closely related viruses such as MPXV and MVA at high specificity, our assay enables distinctions that classical serology alone cannot achieve. More broadly, our results demonstrate that leveraging high-dimensional serological data through ML can reveal latent patterns in immune responses, paving the way for a new generation of seroepidemiological studies with significantly improved resolution. We envision this strategy becoming a blueprint for differentiating antibody immune responses, especially those that result in cross reactive antibody responses to closely related viruses, such as the flaviviruses, encouraging a shift from single-marker assays toward multiplexed, computationally-driven analysis in both research and surveillance contexts.”

Together, these clarifications aim to communicate more clearly how our approach advances the field, not only technically but conceptually, and how it may influence future assay design, data interpretation, and seroepidemiological practice.

- 3) *Following the title “Differentiation between mpox infection and MVA immunization by a novel machine learning-supported serological multiplex assay”, our understanding was that the work consists of the task to (1) differentiate mpox infection and MVA immunization, (2) use multiplexed serology assays, and (3) use ML to extract a meaning model from the data. In the abstract, however, the outcome is described as “our approach enables precise immune profiling and enhances the accuracy and utility of mpox serosurveillance.” To us, there is a disconnect between the purpose of immune profiling (= following multiple immune system components over time) and the need for differentiation between infection and vaccination (= cross-sectional analysis). Or do the authors mean that the versatile antigen panel + ML can be used for both by keeping the antigen panel and tuning the data by ML?*

We thank the reviewer for this thoughtful observation and appreciate the opportunity to clarify the intended scope and implications of our work.

Our primary aim is indeed to differentiate between mpox infection and MVA immunization using a multiplexed serological assay supported by machine learning. We use the term “immune profiling” to refer to the fine-grained, high-dimensional characterization of antibody responses across a panel of poxviral antigens. This multiplexed signal enables machine learning algorithms to detect immunological patterns that support accurate cross-sectional classification.

To demonstrate the robustness of our approach, we trained and tested our models on both acute-phase samples and a longitudinal cohort (epi), showing that the assay can distinguish infection and vaccination status not only in clearly defined cases, but also in more challenging scenarios such as waning antibody responses and post-vaccination breakthrough infections.

We agree with the reviewer that there was a disconnect between the abstract’s phrasing and the core objectives of the work. In particular, our use of “immune profiling” may have unintentionally implied a longitudinal or systems-level goal, which was not our primary focus. Our main goal was instead to establish a robust, cross-sectional classification method.

To clarify this, we have revised the abstract to avoid ambiguity and better reflect the core contribution of the work:

Revised sentence (line 41):

“Integrating ML with high-dimensional serology enables accurate cross-sectional classification of orthopoxvirus immune status and provides a scalable framework for mpox serosurveillance and outbreak preparedness.”

Finally, we acknowledge the reviewer’s insightful question regarding the broader applicability of our approach. While the current study focuses on cross-sectional classification, the versatility of the antigen panel provides a strong foundation for future longitudinal immune monitoring. In observational studies, the assay could be used in its current form to track IgG responses to 15 poxviral antigens over time, following infection, vaccination, or breakthrough infection, and correlate these responses with levels of protection. Thus, our assay could support research questions related to immune durability and response quality, and may prove valuable for profiling immune responses, e.g. to novel orthopoxvirus vaccines.

Revised sentence (line 529):

“Furthermore, the high degree of multiplexing offered by our assay provides a foundation for future studies to longitudinally monitor immune responses to 15 distinct antigens after infection, vaccination, or breakthrough infections.”

We hope this clarification resolves the confusion and better aligns the title, abstract, and aims.

- 4) *Re comment 9) We had previously commented on the absence of evidence about antigen immunogenicity (comment 11) and the authors state that they agree. Still, in the new sentence at line 113ff, in response to comment 9, they state again “To further assess the immunogenicity of the selected recombinant antigens...”.*

Thank you for pointing out this inconsistency. The phrase “To further assess the immunogenicity of the selected recombinant antigens...” was an unintended remnant from an earlier draft and has now been removed as part of streamlining after moving Figure 1 to the Supplementary Information (see our response to Point 14). Throughout the manuscript, we now reserve the term immunogenicity for passages that cite studies directly assessing it (i.e., induction of antibodies after immunization). When referring to our own data, which quantify antigen-specific antibody binding in a multiplex assay, we use reactivity or recognized instead, as appropriate (see Discussion, lines 440 ff., in the tracked-changes document).

- 5) *Re comment 13) We asked why the authors included specific models in their investigation and not why they prioritized GBC in the end. We would have wished for a more focussed investigation and description of their assumptions and then benchmarking these against classic models like lasso or even XGBoost. The authors use random forest (RF) as a more widely used and pre-ML models, which is good, but there is a need to use additional classical approaches and take RF as a baseline to present the improvement in performance of GBC (or LDA), such as in differences of F1 scores. This should be reported also for the sample set used for validation (Table 3).*

We thank the reviewer for the suggestion to clarify the rationale for our model selection and to better contextualize performance improvements.

We focused on Random Forest (RF) and Gradient Boosting Classifier (GBC), as both are widely used, well-understood, and robust models for classification tasks. These models are consistently reported to perform well across biomedical datasets and offer interpretable feature importance rankings.

We considered, but ultimately excluded, additional models for the following reasons:

- XGBoost is conceptually similar to GBC (both are gradient boosting decision tree methods). While XGBoost is optimized for speed and parallel computation, it typically yields results that are very similar to GBC in practice. Our goal was to benchmark representative, well-validated ML models rather than to compare different boosting implementations, so we chose GBC for clarity and integration with our analysis framework.
- Lasso regression is primarily designed for regression and sparse feature selection. It is less commonly applied to multi-class or highly non-linear classification problems, as encountered in our case. Tree-based models like RF and GBC are better suited for capturing the potential non-linear interactions among antigens.

We now explicitly present RF as a baseline model in the Methods, and emphasize incremental performance gains of GBC (and LDA) over RF in both the establishment and validation cohorts. LDA is included for comparison, with an explicit discussion of its distributional assumptions and observed performance.

To further address this point, we have expanded our analysis and revised the manuscript as follows:

- Comparative performance metrics (mean F1 scores and 95% CIs by bootstrap, n = 2000) are now reported for RF, GBC, and LDA, for both the combined acute and epi as well as the validation cohorts.
- A revised Table 3 now summarizes those results.
- We have added a detailed description of the model assumptions and rationale for model selection in the Methods part of our manuscript.

Revised sentence (line 686):

“Six models were trained using three algorithms, Random Forest (RF), Gradient Boosting Classifier (GBC), and Linear Discriminant Analysis (LDA), on IgG alone or combined IgG/IgM data from the acute cohort, the epi cohort, or both combined (see Fig. 1). We selected RF and GBC to represent two common, complementary ensemble learning approaches: bagging and boosting, respectively. These models are consistently reported to perform well across biomedical datasets and offer interpretable feature importance rankings⁷¹⁻⁷³. RF served as a strong baseline model due to its stability and interpretability, while GBC was chosen for its ability to capture complex, non-linear interactions, a key requirement in distinguishing overlapping serological profiles.”

- 6) *Before such an exercise, we suggest that the authors revisit all the fundamental requirements for the chosen models. As they write in lines 334ff about the distribution of the data as a criterion for the models, it is necessary that the model assumptions match with the actual data. Most models are not specifically built for serology data but can we really expect a normal distribution in immune response data (or would forcing a distribution onto the data bias the outcome)? We would rather expect the data to be skewed by tails indicating infections (upwards) or waning (downwards), or even observe bimodal distributions (\pm vaccination). Hence, we have some doubt that these aspects have been taken into account to a degree necessary for the robustness of the ML claims.*

We thank the reviewer for highlighting the importance of model assumptions in relation to data distribution, a point that is particularly relevant in serological datasets, which often exhibit skewed or multimodal distributions due to variable infection timing, vaccination status, and waning immunity.

We fully agree that Linear Discriminant Analysis (LDA) relies on assumptions of normally distributed features and equal covariance matrices across classes. As noted in the manuscript (lines 486 ff), this likely contributed to LDA’s reduced performance in differentiating MPXV breakthrough infections from MVA-only vaccinated individuals. We have now explicitly discussed this limitation in the revised Discussion section, highlighting it as a potential reason for misclassification in immunologically complex samples.

Revised sentence (line 490):

“Such assumptions are unlikely to hold in serological datasets, which frequently exhibit skewed or bimodal distributions driven by factors such as vaccination history, breakthrough infections, and waning antibody responses. These distributional characteristics likely contributed to the reduced performance of LDA, particularly in distinguishing complex or overlapping immune states. “

To further support this point, we have:

- Added density plots of antigen-specific IgG distributions across serogroups (Pre, MVA, MPXV) to the Supplementary Information (Fig. S12). These show that all features indeed exhibit non-normal, skewed, or bimodal distributions, consistent with the reviewer’s expectations.
- Reiterated in the Methods that non-parametric tree-based models (RF and GBC) were included specifically because they do not rely on distributional assumptions, making them better suited for this type of serological data.

Revised section (line 689):

“We selected RF and GBC to represent two common, complementary ensemble learning approaches: bagging and boosting, respectively. These models are consistently reported to perform well across biomedical datasets and offer interpretable feature importance rankings⁷¹⁻⁷³. RF served as a strong baseline model due to its stability and interpretability, while GBC was chosen for its ability to capture complex, non-linear interactions, a key requirement in distinguishing overlapping serological profiles. Given the non-normal, often skewed feature distributions (Fig. S12, Shapiro–Wilk tests), we prioritized tree-based models (RF, GBC) that capture non-linear interactions; LDA served as a linear baseline to benchmark performance under stronger distributional assumptions (multivariate normality and equal covariance across classes), assumptions that are often violated in immune response datasets due to skewed, heterogeneous, or bimodal antigen distributions. XGBoost was not included to avoid redundancy with GBC. Lasso regression was considered less appropriate due to its limited utility in multi-class settings and assumption of linear separability.”

We appreciate this comment, as it helped us strengthen the justification for our model choices and clarify the limitations of classical models in this context.

- 7) As Figs 2 + 4, we understand the idea behind evaluating, training, and testing the models in the acute and epi sample sets interchangeably, but remain uncertain about what the differences are between the models. Do they include a different sub-panel of the antigens or different weights?

We thank the reviewer for this question. To clarify: all models were trained and evaluated using an identical set of antigens and serum samples; there were no differences in input variables, sample selection, feature selection, or weighting between models. The only distinction among models lies in their respective learning algorithms (e.g., Random Forest, Gradient Boosting, LDA).

All models were evaluated using stratified 5-fold cross-validation, repeated three times, ensuring that each model experienced the same data splits. We have added explicit clarification in the Methods section stating that observed performance differences are attributable solely to model architecture and not to differences in input data.

Revised section (line 700):

“All machine learning models were trained using the same set of antigen features and serum samples, with no variation in input data across models. To evaluate model performance and robustness, we applied stratified 5-fold cross-validation, repeated three times (15 runs per model). This approach ensures that each model is exposed to the same training and testing conditions, allowing direct comparison of classification performance based solely on differences in model architecture.”

- 8) Also, we need to know more clearly what the added accuracy of this study is over less elaborated approaches, such as an ELISA assay based on two proteins. Without such reference, it remains unclear whether the approach taken is necessary. That said, the ML-based analysis seems to demonstrate that some improved classification performance can be achieved compared to using the single antigen ATI-N, but what about using two variables: E8 and ATI-N (see Fig. 2b-c, and main antigen branches of Fig. 5a)?

We thank the reviewer for highlighting the need to benchmark our ML-based approach against less elaborate, dual-antigen methods such as an ELISA based on ATI-N and E8. In response, we explicitly compared the performance of our ML model to threshold-based classifiers using both antigens individually and in combination, as detailed in Table S13.

In the validation cohort, the combined ATI-N/E8 classifier achieved a mean F1-score of 0.65 (95% CI: 0.57–0.72), while the ML (GBC) model achieved 0.70 (95% CI: 0.63–0.77). A larger, practically relevant difference was observed in the combined acute and epi cohort, used to establish the model (ML: 0.82 [0.80–0.85] vs. ATI-N/E8: 0.72 [0.70–0.75]). These differences were consistent for both binary and multiclass classification, with the greatest improvements seen in multiclass settings and larger datasets. In the independent validation cohort, the improvement did not reach statistical significance due to the wider confidence intervals, but the direction was consistent.

We extended the comparison to the second-best single antigens for differentiation between MPXV infected and non-infected sera (ratio A35-MPXV/A33-VACV) and for differentiation between seropositive and seronegative (B6-MPXV). Here, the second-best single antigens performed significantly worse in classifying sera as MPXV infected as well as in multiclass settings in the independent validation cohort as compared to the GBC model.

These results demonstrate that ML-based multiplex serology yields a robust and significant increase in classification accuracy that cannot be achieved by simply combining two antigens. We have added these analyses to the revised manuscript (Table S13).

While classical statistical approaches could theoretically combine more than two antigens, this quickly necessitates algorithms capable of integrating complex antibody signatures, precisely what our ML approach is designed to do. For transparency, we included LDA as a classic multivariate classifier, but as discussed above, its performance is limited by distributional assumptions not met by our data. Additionally, incorporating more antigens rapidly increases the efficiency of multiplex assays, significantly reducing workload, reagent, antigen, and sample consumption compared to conventional ELISAs.

We trust that this additional analysis and transparent comparison clarify the necessity and added value of the ML-based approach as compared to less elaborate dual-antigen assays.

Specific changes and relevant sections in the revised manuscript:

Revised section (line 365):

“Previous studies used classifiers based on single or few antigens to distinguish MPXV infection from other sera or to identify orthopoxvirus seropositivity resulting from infection or vaccination^{34,35}. To directly quantify the performance gain achieved by our ML-based approach, we systematically compared our GBC model (baseline, without classification enhancement) against simplified single-, dual-, or even triple-antigen classifiers on the same cohorts. This head-to-head evaluation was essential, as comparisons based solely on published performance metrics can be misleading due to differences in cohort composition, sample size, and sampling timepoints.

Using ROC analyses to determine threshold values and to compare the performance of single antigens for distinguishing seropositive from seronegative samples, E8-MPXV emerged as the best single antigen, followed by B6-MPXV, whereas ATI-N-CPXV and the A35-MPXV/A33-VACV binding ratio best differentiated MPXV-infected from MVA vaccinated samples (Fig. S11, threshold values and performance Table S12). For multiclass classification, we evaluated rule-based schemes using pairs of top-performing antigens to approximate the ML classes (pre-immune, MVA, MPXV): (i) ATI-N-CPXV + E8-MPXV and (ii) the A35-MPXV/A33-VACV binding ratio + B6-MPXV. Using ROC-derived thresholds,

samples were assigned as follows: MPXV if ATI-N-CPXV positive (or ratio positive); MVA if seropositive (E8-MPXV or B6-MPXV positive) but ATI-N/ratio negative; pre-immune if both markers were negative.

Across all tasks, the GBC-based ML approach consistently outperformed simplified classifiers (Table S13). While binary serostatus classification using single antigens approached ML performance (F1 score decrease of only 0.02–0.05), classification of MPXV-infected sera and multiclass predictions showed markedly poorer performance (F1 score decrease of 0.12–0.20). Importantly, reliance on second-tier antigens, such as the A35-MPXV/A33-VACV ratio, resulted in a substantial performance drop on the validation panel, underscoring the unique value of ATI-N-CPXV as a single discriminator of MPXV-infections.“

- 9) Again, we remain skeptical that an improvement of this magnitude justifies the use of such a broad antigen panel and complex analytical approach in a real-world setting, even in epidemiological studies. As stated in lines 280ff, “Among all antigens, ATI-N-CPXV performed best in distinguishing MPXV-infected individuals, achieving an F1 score of 0.83. Binary classification using GBC achieved slightly better results, with an F1 score of 0.85 and higher specificity (0.95 vs. 0.91 for ATI-N-CPXV alone; Table S13).” We really appreciate this transparency, but it is still unclear to us if a F1 score gain of 0.02 is striking enough to justify the efforts.

We appreciate the reviewer’s thoughtful critique regarding the practical value of a 2 percentage point gain in F1 score for binary classification between MPXV-infected and non-infected individuals using ATI-N-CPXV alone versus the GBC model. We agree that, taken in isolation, this modest improvement may not appear sufficient to justify the additional complexity.

However, as detailed in our expanded analyses (see response to point 8 and Table S13), the key advantages of the ML-based multiplex approach become more apparent in two critical scenarios: (i) multi-class classification tasks, and (ii) analyses on the larger, more heterogeneous combined acute and epi cohort. In these contexts, the performance gains of GBC over single or dual-antigen classifiers are substantially larger, and frequently cross thresholds that are widely regarded as meaningful in diagnostic assay development.

Moreover, we believe that the robustness of the ML approach offers a crucial benefit beyond marginal increases in F1 score. By leveraging patterns across a panel of individually moderate or weak antigens, the ML model can maintain high classification performance, even when the most discriminatory single antigens (ATI-N-CPXV, E8-MPXV, D8-VACV) are excluded from the panel. Our new benchmark study demonstrates that removing these top contributors from the model results in only a minor drop in F1 score, underscoring the resilience and redundancy provided by multiplexed, ML-based analysis.

Revised section (line 389):

„In contrast, our GBC model demonstrated remarkable robustness. Removing ATI-N-CPXV, E8-MPXV, or D8-VACV individually, or even ATI-N-CPXV together with E8-MPXV, had a negligible effect on classification performance (F1 decrease \leq 0.01–0.02 across combined and validation cohorts; Table S14). Only exclusion of all three antigens reduced performance modestly (F1 decrease = 0.04 on the combined cohort; F1 decrease = 0.01 on validation). Thus, our ML-based approach not only outperforms simpler single- or dual-antigen strategies but also provides greater robustness and redundancy, enabling stable classification even when key antigenic markers are removed.“

Revised section (line 460):

“Across binary and multiclass tasks, ML-based classification consistently outperformed single-, dual-, and triple-antigen rules evaluated on the same sera. Even when restricted to the strongest single markers, ML, particularly GBC, achieved higher accuracy for identifying MPXV infections, the endpoint most relevant for seroepidemiology. These gains indicate greater robustness to confounders (e.g., residual smallpox immunity) and improved transferability to independent cohorts compared with rule-based approaches. Remarkably, our benchmarking demonstrates that the performance of the top-performing ML algorithm (GBC) was largely unaffected when ATI-N-CPXV or even ATI-N-CPXV together with E8-MPXV, the two most influential antigens, were excluded from the training and testing dataset. This finding indicates that the remaining antigens provide sufficient redundancy to capture key patterns in the antibody response, enabling reliable discrimination between MPXV-infected, MVA-immunized, and pre-immune sera even in the absence of the strongest antigens. These results underscore the added value of ML-based multiplex serology over classical single-antigen approaches, highlighting not only the robustness of the ML method, but also its ability to maintain high classification performance even when major antigenic markers are missing.”

This finding is particularly relevant for real-world and epidemiological applications where antigen performance may vary across populations or over time. It demonstrates that the ML model is not dependent on a single “star” antigen, but rather can integrate weaker, partially overlapping signals to achieve robust group discrimination, a capability that classical, threshold-based approaches fundamentally lack.

In summary, while single-antigen classifiers like ATI-N-CPXV remain attractive for their simplicity in straightforward binary tasks, our ML approach provides both significant performance advantages in more complex or heterogeneous settings and unique robustness against loss or variability in individual antigen performance. We believe these features justify the added complexity in contexts where high accuracy, flexibility, and resilience are critical.

- 10) We also remain uncertain about the necessity of the multiplexed serology over single or dual analyte assays, and how others should use the presented findings. In lines 353ff, the authors suggest that “Future work should consider incorporating additional antigens with higher specificity and explicitly stratifying models by breakthrough infection status and childhood immunization history.” To us, this reads as if the current approach is still insufficient, and better antigens and assays should be designed. There is also a need to reference the outcome of this study to other efforts attempting to achieve a similar outcome.

We thank the reviewers for the opportunity to communicate the strengths of our approach more clearly, enabling readers to better understand its potential for improved diagnostics, its advantages over single- or dual-analyte assays, and how it compares to other efforts in the field.

First, we would like to clarify that we consider our current approach to be robust and well-validated, offering a distinct advantage over existing assays. Nonetheless, we recognize that any diagnostic method can be further refined—for example, by incorporating additional antigens. In highlighting these areas, our intention was to indicate possible avenues for future optimization rather than to imply that such modifications are necessary or would definitively enhance the current assay. We appreciate that our original wording may have caused confusion on this point. Accordingly, we have

revised “Future work should consider” to “Future work could consider” to better reflect our intention.

Second, we believe that our responses to Points 8 and 9 now more clearly highlight the advantages of our combined multiplex-serology approach in conjunction with ML-based analysis, particularly in comparison to single- or dual-analyte assays.

Third, in our initial submission, we provided a comparison of our assay’s performance with that of previously published assays. To streamline the manuscript, we removed this section in the revision. However, we recognize that direct numerical comparisons are complicated by confounding factors such as differences in antigen selection, sample size, timing post-infection, and composition of negative serological cohorts (e.g., vaccination history, population risk). These factors can significantly impact reported assay performance and make direct comparisons potentially misleading. Hence, our benchmark comparisons against simpler classifiers best demonstrate the performance gain of our approach. We now also report data on sensitivity and specificity, specifically for correctly classifying sera as MPXV-infected, as this is the most relevant use case for seroepidemiological studies. This facilitates a direct comparison of our assay’s performance with values reported in the literature.

To address this point comprehensively, we have reinstated the previously reported numbers and extended them with data from newly published assays that have appeared since our preprint went online.

Revised section (line 409):

“Specifically, depending on the assay and setup, previously reported sensitivities and specificities in the literature vary, ranging from 93%/98% for a combination of mpox-specific peptides and recombinant proteins⁴⁶, 86%/90% for an ELISA with MPXV-specific peptides⁴⁷, 86%/100% for a post-absorption ELISA³⁸, 100%/90% for another peptide-specific ELISA and 92-100%/100% for a combination between titers to VACV/MPXV binding and titer decline⁴⁸. Finally, recently published multiplex assays similar to our approach achieved sensitivities/specificities of 88%/97% and 89%/80% for differentiating MPXV-infected subjects from MVA-vaccinated, respectively^{34,35}. However, confounding factors such as differences in antigen selection, sample size, timing post-infection, and the composition of negative cohorts (e.g., vaccination history, population risk) can substantially influence reported assay performance, rendering direct comparisons potentially misleading.

Despite these limitations, our assay compares favorably with prior work, yielding 86–94% sensitivity and 88–93% specificity for distinguishing MPXV infection from both pre-immune and MVA-vaccinated sera, depending on the cohort and whether low-confidence predictions were excluded. Crucially, the best-performing model (GBC) consistently outperformed single-, dual-, and triple-antigen rule-based classifiers across binary and multiclass tasks in both the combined dataset and the independent validation cohort. These findings demonstrate a clear performance advantage of ML-based multiplex serology over simplified approaches.

Furthermore, compared to previously published studies, our work is distinguished by the large number of sera analyzed, the inclusion of a comprehensive panel of 15 unique antigens, measurement of both IgG and IgM responses, and evaluation of both acute and convalescent phase sera. Moreover, our results are validated using an independent cohort and include samples from individuals with a history of smallpox vaccination.”

Finally, regarding the question of how others might use the findings presented here, we believe that our work advances the field in several important ways. By including benchmark studies on the performance of smaller antigen panels, we have identified the optimal combination of antigens that

achieves high performance while reducing assay costs (see our response to points 9 and 11). Our comparison of different algorithms demonstrates that GBC achieves the best results, providing a strong starting point for the development of similar assays. We now also include threshold values for single antigens (Table S12), that can be used by others, along with information about the best performing antigen to establish assays based on single antigens. Furthermore, by making all data and code publicly available, we offer a resource that others can use to train and test their own machine learning algorithms.

Collectively, we believe that these contributions, along with our emphasis on including diverse epidemiological cohorts for robust performance, offer significant new insights that can be adapted to develop similar assay or data analysis pipelines, not only for mpox immune response differentiation, but also for other closely related virus families, as discussed elsewhere in the manuscript.

- 11) It remains to be seen if dual detection of IgG and IgM is needed for all studies. While the high-throughput platform allows testing of many antigens across many samples with minimal sample consumption, it remains a costly setup. It would be valuable to know whether the authors have assessed the individual contribution of each antigen to the model and whether a reduced panel could achieve similar performance.

Thank you very much for the suggestion to include an assessment of the individual contribution of each antigen. As already partly answered under point 9, we have now performed a thorough recursive feature selection to identify a minimum panel of antigens, that could perform the classification task with similar performance as compared to the full panel of antigens.

Revised section (line 266):

“To quantify the contribution of individual antigens to model performance, we performed recursive feature elimination on the GBC model (Fig. S9). Including all antigens yielded the highest performance (baseline $F1 = 0.82 \pm 0.02$ for IgG/IgM data), but five antigens could be removed with only a minor reduction in accuracy ($F1 = 0.81 \pm 0.02$). Performance began to decline after removing ATI-C-CPXV and A33-VACV, and more markedly with the exclusion of Delta-VACV ($F1 = 0.77 \pm 0.01$). The largest drops occurred when A35-MPXV, ATI-N-CPXV, and D8-VACV were removed, leaving E8-MPXV as the strongest single antigen ($F1 = 0.50 \pm 0.02$ for IgG; 0.62 ± 0.03 for combined IgM/IgG). These analyses indicate that a minimum panel of eight antigens (ATI-N-CPXV, ATI-C-CPXV, D8-VACV, E8-MPXV, A33-VACV, A35-MPXV, VACV lysate, and Hep-2 lysates, comprising Delta-VACV) achieves performance comparable to the full panel.”

Revised section (line 514):

“Conversely, we demonstrate that high classification performance of the GBC algorithm can be maintained with a reduced panel of just 8 antigens. Selecting the most informative antigens enables minimization of multiplex assay costs while preserving, and potentially optimizing, overall assay performance.”

With regard to IgG and IgM detection, our results demonstrate that including IgM measurements improves overall assay performance. Nevertheless, depending on the serological cohort, IgG alone often provides sufficient discrimination. As noted in the manuscript, newer generations of Luminex instruments can simultaneously quantify IgG and IgM, so capturing both is possible without

additional measurements on such platforms. In resource-limited settings, it remains feasible to measure only IgG. Our work offers robust benchmarks for the expected performance loss in this scenario and can guide decisions about whether this trade-off is acceptable. We have included this information also in the discussion of the revised manuscript:

Revised section (line 520):

“Advances in assay technology, such as simultaneous IgG and IgM detection using the latest Luminex platforms⁶², may further streamline and enhance the assay’s diagnostic utility. Our results demonstrate that including IgM measurements improves overall assay performance. Nevertheless, depending on the serological cohort, IgG alone often provides sufficient discrimination. In resource-limited settings, it remains feasible to measure only IgG. Our work offers robust benchmarks for the expected performance loss in this scenario and can guide decisions about whether this trade-off is acceptable.”

- 12) Re comment 16) We had asked to use the term “panel” only for the assay and set of analytes used, not to mix it with and confuse it with the different sets of samples used. However, there remains inconsistency in adapting those changes to more clearly differentiate assay and sample aspects (including the figures).

Thank you very much for bringing this to our attention. We apologize for any confusion caused by the inconsistent use of terminology. We have now thoroughly revised all figures, tables, and text to eliminate any remaining inconsistencies. The term “panel” is now used exclusively to refer to antigens, “serological cohort” is used to distinguish between the different sample sets (“acute,” “epi,” [referred to in combination as “combined”] and “independent validation”), and “group” denotes the three prediction groups (Pre, MVA, MPXV). We believe these changes will ensure clear differentiation between assay components and sample aspects throughout the manuscript.

- 13) The message of Fig 3 is unclear to us and the use of the term panel instead of study sets adds confusion. What is the difference between the three double-boxes in the parameter section - aren’t they using the same assay panel for all sample sets?

To resolve this confusion, we revised former Fig. 3 and now present it as new Fig. 1. We clarified the distinction between the assay antigen panel, the serological cohorts, and the IgG vs. IgG/IgM datasets, and how each was used during ML training and testing, to guide the reader from the outset and avoid confusion in terminology, assay setup, and methodology.

To answer the question directly: the multiplex assay was applied to all serological cohorts, and both IgG and IgM were measured, yielding one comprehensive dataset. ML algorithms were then trained on the acute, epidemiological, or combined datasets, using either IgG-only data or the combined IgG/IgM data. We hope the new Fig. 1 clearly communicates the workflow and helps resolve this confusion.

- 14) Re comment 19) The new paragraph clarifies the quality control work. This paragraph could be included in the method sections or in results, maybe in a shorter and modified version to fit the section’s purpose. However, to show the reproducibility of the measurement, we would rather suggest checking the level of agreement of the 2 experiments in classifying

positive and negative control samples. In Fig. 1, which could also be relocated to the supplementary to keep a focus, the data clouds are rather narrow for some of the antigens, making it difficult to appreciate the actual magnitude of the variation between the two different experiments on different antigens. Antigens like ATI-N, B6 and E8 for instance, show a number of samples visibly deviating from the trajectory.

Thank you very much for your valuable comments and suggestions regarding the assessment of measurement reproducibility and quality control.

In response, we have incorporated a dedicated section in the Methods and provided the corresponding quality control data as a new supplementary figure (Figure S12).

To strengthen our analysis, we extended the number of duplicate measurements by including an additional independent quantification of 39 sera from the epidemiological (epi) cohort. In the revised figure, we now explicitly quantify agreement between the two independent measurements for each antigen by calculating the coefficient of determination (R^2) from linear regression, thereby providing a quantitative metric for reproducibility on the continuous scale. We acknowledge that the data clouds for some highly reactive antigens appear narrow due to limited variability, but the inclusion of R^2 and F1 scores now quantifies the level of agreement and makes outliers and deviations (e.g., for ATI-N, B6, E8) more readily appreciable.

As further recommended by the reviewers, we also evaluated the consistency of classification into seropositive and seronegative samples between experiments. For each antigen, sera were classified based on cut-off values derived from method comparison with IFA (titers $\geq 1:320$ defined as seropositive), and F1 scores were calculated to assess the agreement in classification across measurements.

We believe that these additional metrics offer a more transparent and comprehensive assessment of both overall reproducibility and instances of greater variability, as highlighted by the reviewer. We thank you again for prompting us to strengthen this aspect of the study.

Revised section (line 659):

“Assessment of measurement reproducibility

Reproducibility was assessed by independently quantifying antibody levels in 168 serum samples (39 from the epidemiological cohort, 129 from the validation cohort). For each antigen, the coefficient of determination (R^2) was calculated by linear regression between the two independent measurements. Additionally, the F1 score was determined for each antigen to evaluate the agreement in binary classification as seropositive or seronegative. High reproducibility was observed for highly reactive antigens, as indicated by R^2 values and F1 scores approaching 1 (see Figure S13). Agreement was generally higher for IgG than for IgM measurements, while antigens with lower immunoreactivity exhibited reduced concordance.”

Furthermore, as suggested, we have moved the former Figure 1 to the Supplementary Information (Fig. S1) to maintain the focus on the ML comparison.

- 15) Re comment 26) We asked for a cut-off value that can be used by others. It seems the term is used here solely for the model and not as a value that can be adapted elsewhere (e.g. MFI > 2000 or [c] > 1 ng/ml ...). In this regard, we found that the authors have invented terms like “Multiplex Delta Quant” (Fig 1) instead of simplifying the message in units that are

translatable to other studies (like $\mu\text{g/ml}$ or else). This continues in Fig 2 (Min Max norm.). It remains unclear to us how others could use these features in their investigations.

Thank you for raising this important point. We appreciate the opportunity to clarify our terminology and further enhance the practical value of our results for the field.

To address your comment on the term "Multiplex Delta Quant," we have now included a brief explanation in the legend of Figure 1 (now Fig. S1), stating explicitly that these values represent quantified IgG concentrations (ng/mL), using Vaccinia immune globulin (VIG) as a reference standard. While this information was provided in the Methods section, we agree that highlighting it directly under the figure will aid clarity for readers.

Regarding the min-max normalized data in Figure 2, normalization was performed to align with best practices in machine learning, as all input features were standardized before model training. We present the data on this normalized scale to reflect the input used by the ML algorithms and to facilitate interpretation of feature importance across antigens.

In addition, we now provide explicit threshold values, expressed as \log_{10} IgG concentrations (ng/mL), for the best-performing antigens in Supplementary Table S12. Because all underlying data and scripts are openly available, readers can derive additional cut-off values as required.

We did not provide thresholds in MFI units, as MFI values are not directly transferable between laboratories due to substantial inter-lab variability, background effects, and lot-to-lot differences. Best practice, therefore, is to relate quantitative results to a defined standard. In the absence of a WHO-certified international reference, we used VIG, which is no longer available from BEI Resources. While we retain sufficient VIG for internal use, sharing is restricted by material transfer agreements, and no broadly available alternative standard currently exists. We think that development of an international harmonized standard would be highly valuable for future serosurveillance efforts, yet well beyond the scope of our manuscript.

16) Re comment 28) We refer to other viral infections or vaccinations (e.g. influenza) that could influence the data due to molecular mimicry or assay cross-reactivity.

Thank you very much for this clarification, which now allows us to address this point more directly. It is well established that people with HIV (PWH) are at higher risk of more severe outcomes following mpox infection, particularly if HIV infection is untreated or immunosuppression is advanced (low CD4 counts, high viral load; [https://doi.org/10.1016/S0140-6736\(23\)00273-8](https://doi.org/10.1016/S0140-6736(23)00273-8)). For those on effective antiretroviral therapy, outcomes are generally similar to those without HIV.

MVA-BN vaccine effectiveness also appears to be somewhat reduced in PWH, as shown by Hillus et al. (2025), but breakthrough infections in vaccinated individuals were generally milder, indicating that vaccination still offers meaningful protection (Hillus et al., 2025, *Lancet Infect Dis*, "Safety and effectiveness of MVA-BN vaccination against mpox in at-risk individuals in Germany," [https://doi.org/10.1016/S1473-3099\(25\)00018-0](https://doi.org/10.1016/S1473-3099(25)00018-0)).

There also appears to be an impact of HIV status on the rate of seroconversion following MVA-BN vaccination, with lower seroresponse rates in PWH, as reported by Liu et al. (2025) in the *International Journal of Infectious Diseases* ("Short-term evolution of Mpox-specific IgG and neutralizing antibodies among individuals undergoing MVA-BN vaccination," <https://doi.org/10.1016/j.ijid.2025.107830>).

To clarify this point, we have added the following section to the Discussion:

Revised section (line 511):

“Explicitly considering HIV status in future studies would enable investigation of the humoral immune response in people with HIV. This population is at increased risk for mpox infection and may experience different clinical outcomes or reduced effectiveness of MVA vaccination^{60,61}. However, vaccination has been shown to significantly reduce the severity of symptoms in breakthrough infections.”

To our knowledge, aside from prior exposure or vaccination with other orthopoxviruses (e.g., smallpox, vaccinia, cowpox), there are no published reports indicating that common viral infections or vaccinations result in serological cross-reactivity or molecular mimicry that would impact mpox/orthopoxvirus antibody assays. Our antigen panel was selected for orthopoxvirus specificity, and all assays included negative control samples from individuals with diverse backgrounds. Explicitly, our negative control samples were partly positive for measles, mumps or rubella. In addition, Otter et al. (Nat Commun, 2023) explicitly tested for cross-reactivity with sera from individuals with confirmed CMV, EBV, and VZV infections, as well as rheumatoid arthritis, and found no significant binding to orthopoxvirus antigens in these groups. This indicates that common herpesvirus infections and autoimmune backgrounds do not impact the specificity of orthopoxvirus serology assays.

- 17) We also noted that the authors do not use any detergent in their assay buffer. Can they please justify why they omitted these very common additives? Is there no concern about unspecific binding?

Thank you for this important question. Our protocol follows the recommendations in the xMAP Cookbook (4th Edition) and standard commercial Luminex serology kits, which specify PBS with 1% BSA as the assay buffer without added detergent. While detergents such as Tween 20 are commonly used in ELISA buffers to further suppress nonspecific binding, the Luminex protocol relies on high BSA concentration in the assay buffer and the use of Tween 20 in all wash steps (PBS + 0.1% Tween 20). This approach is empirically established for bead-based multiplex assays, possibly to avoid interference with bead–antigen interactions or multiplex detection. We also perform stringent intermittent washing steps using Tween-containing buffer to ensure that nonspecific binding is minimized. To date, we have observed no issues with background binding, and our negative controls remain consistently low, supporting the effectiveness of this approach in our hands.

- 18) Re comment 30) We have also requested to avoid relative terms, like high or low, but already in the abstract, we came across “... independent cohort (n = 143) maintained high classification ...”. If we read correctly, the F1 score decreased to 0.77 (Table 3), and this should be stated.

Thank you very much for bringing this to our attention. We carefully revised the complete manuscript and specified all relative terms with concrete numbers. The abstract was also revised and the numbers are now clearly stated. We now focus on the reporting sensitivities and specificities for detection of MPXV infections, as those numbers are important to be able to compare our assay with assays previously published (also in response to point 10), but also report full numbers.

Revised sentence (line 39):

“In an independent validation cohort (n = 143), GBC (F1 = 0.70) robustly detected MPXV infections, including breakthrough cases, with 92% sensitivity and 88% specificity. “

We hope these comprehensive revisions address all outstanding concerns, but are of course happy to further clarify or adjust the manuscript if needed.

Point-by-Point Reply

We thank all reviewers for their insightful, fair, and thorough evaluation of our manuscript. We are pleased that our revisions have clarified the remaining issues and further sharpened our key message. We are also glad that all major concerns raised by the reviewers have been fully resolved, with only minor points remaining.

We have carefully addressed all remaining issues and editorial requests.

REVIEWER COMMENTS

Reviewer #4 (Remarks to the Author):

We thank the authors for their continuous efforts to improve the manuscript and follow our constructive suggestions.

Overall, the manuscript has substantially improved in text, flow, reasoning, and clarity of the figures.

The benefit of the approach has become a lot clearer. Especially now there is a thorough and transparent comparison with classical methods and their performance (lines 301 ff + Tables S12 + S13). As stated by the authors, AI can support multi-analyte serology for multi-class categorization efforts. Hence, the presented framework showed greater resilience and outperformed classical singular/dual antigen concepts in the presented data. We can foresee an interest in the framework as a mathematical exercise and opportunity for benchmarking in real-life scenarios of other infectious disease phenotypes that might remain more challenging to disentangle (e.g. flaviviruses).

We appreciate the positive assessment regarding the substantial improvement of the manuscript in text, flow, reasoning, and figure clarity. We are pleased that the added comparison with classical methods helped illustrate the advantages of our approach. We are grateful that the review process helped us refine and highlight the scientific value of our work.

As closing remarks that can be handled with the editorial team, we wish that the authors:

- 1) modify the title to better reflect these key aspects. One suggestion could be: Machine learning-supported framework for the classification of mpox infection and MVA immunization from multiplexed serology data.*

We thank the reviewer for this excellent suggestion. We agree that the proposed title captures the essence of our manuscript and have adopted it as the new title.

- 2) add units to all figures presenting data irrespective of the model used. Stating the arbitrary unit [AU] or else will assist future readers to better connect with the presented data.*

We have added units to all figures in both the main manuscript and the Supplementary Information. Units were omitted only for dimensionless values (e.g., ratios or normalized data). Explanations have been added in the respective figure legends where appropriate.

We believe that the revised manuscript now fully addresses all editorial requirements and reviewer feedback and is ready for publication in Nature Communications.

With kind regards,
Daniel Stern (on behalf of all authors)